# Detrital zircon provenance record of the Zagros mountain building from the Neotethys obduction to the Arabia-Eurasia collision, NW Zagros fold-thrust belt, Kurdistan region of Iraq

Renas I. Koshnaw[1*], Fritz Schlunegger[1], Daniel F. Stockli[2]

[1]Institute of Geology, University of Bern, Baltzerstrasse 1+3, CH- 3012 Bern, Switzerland
[2]Department of Geological Sciences, Jackson School of Geosciences, University of Texas at Austin, Austin, TX 78712, USA

*Correspondence to*: Renas I. Koshnaw (renas.i.koshnaw@gmail.com)

*Present address: Department of Structural Geology and Geodynamics, Geoscience Center, University of Göttingen, Goldschmidtstraße 3, 37077 Göttingen, Germany

**Abstract.** Recognition of a new angular unconformity and a synthesis of new detrital zircon U-Pb and (U-Th)/He
provenance records, including zircon (U-Th)/(He-Pb) double dating, from the NW Zagros elucidate the basin dynamics of the foreland wedge-top and intermontane units, as well as the tectonic processes in the source terranes in response to the different geodynamic phases. In this contribution, we present field observations and detrital zircon provenance data from hinterland basins to reconstruct the basin dynamics and the underlying tectonic controls in the NW Zagros in the Kurdistan region of Iraq. Results reveal that the deposition of the suture zone units of the Red Beds Series (RBS; Suwais Group,
Govanda Formation, Merga Group) occurred in an intermontane basin on top of folded Upper Cretaceous units, and that the RBS deposits rest with an angular unconformity on the underlying older strata. The RBS provenance data point to the Paleogene Walash-Naopurdan-Kamyaran (WNK) complex as a source area and imply a substantial decrease of magmatism by ~36 Ma, as reflected by the youngest age peaks. New detrital zircon provenance data from the hinterland wedge-top units of the proto-Zagros foreland basin (the Tanjero, Kolosh, and Gercus Formations) exhibit exclusive derivation from the
Upper Cretaceous Neotethys ophiolitic terranes, which differs from the provenance of the older Lower Cretaceous and Paleozoic units that are dominated by the Paleozoic and Neoproterozoic age spectra. These shifts in provenance between different tectonostratigraphic units argue for a sediment source reversal from the W to the E in response to ophiolite obduction, arrival of the WNK complex, and commencement of the Arabia-Eurasia continental collision during the latest Eocene (<36 Ma). According to the provenance data, the incipient collision was followed by the deposition of the RBS in the
hinterland of the proto-Zagros fold-thrust belt, and the connection of drainages with the collision-related Neogene foreland basin.


# 1 Introduction

In sedimentary basins adjacent to collisional mountain belts, synorogenic clastic deposits potentially provide information on the geodynamic setting prior to and during the collisional processes. Understanding the causes and consequences of mountain building processes is critical for the reconstruction of paleogeography, paleoclimate, and for the understanding of the long-term deformation and erosion history (Molnar and England, 1990; Avouac and Burov, 1996; Najman et al. 2010, Norton and Schlunegger, 2011; Avouac et al., 2015). Hinterland basins, such as wedge-top and intermontane basins are valuable archives for the assessment of the exhumation and unroofing history of the adjacent uplifted terranes because of their proximity to the source areas. Nevertheless, well-preserved ancient stratigraphic successions are scarce due to the deformation of sedimentary strata as orogenesis proceeds (Horton et al., 2012; Orme et al., 2015). A possible approach to overcome this drawback is the utilization of the geochronologic and thermochronologic records, which are preserved by detrital zircons (e.g. Cawood et al., 2012; Webb et al., 2013; Gehrels, 2014; Colleps et al., 2020).

The Zagros fold-and thrust belt is a prominent collisional orogen on Earth that stretches for ~2000 km across the Middle East (Fig. 1). This orogenic belt formed during the Late Cretaceous and the Cenozoic as a consequence of the convergence and the subsequent collision of the Arabian and Eurasian continental plates (Dewey et al., 1973; Hempton, 1985; Dercourt et al., 1986; Mouthereau et al., 2012). This prolonged history of deformation resulted in an amalgamation of deferent tectonic terranes that were situated between the Arabian and Eurasian plates, such as Bisotoun block, the middle Cretaceous intraoceanic oceanic subduction and back-arc spreading zone, the early Tertiary magmatic domain. Because the accretion processes overprinted the preceding tectonic configurations (Wrobel-Daveau et al., 2010; Agard et al., 2011; Vergés et al., 2011; Barber et al., 2018, 2019), any paleotectonic models particularly for the NW Zagros suture zone have been associated uncertainties. Previous paleotectonic reconstructions in the NW Zagros suture zone involved several subduction zones, a mantle plume, and different loci for specific tectonic terranes and sedimentary successions of suture zones (e.g. Numan et al., 1997; Karim et al., 2008; Al-Qayim et al., 2012; Azizi et al., 2013; Whitechurch et al., 2013; Ali et al., 2014, 2019; Moghadam et al., 2020). Similarly, possibly due to the complexity of the convergence history, various timings have been suggested for the collision (e.g. Zhang et al., 2016, Fig. 2) with a conceivable minimum age during the late Oligocene (Koshnaw et al., 2019).

This research aims at constraining the basin dynamics recorded by the suture zone deposits and the wedge-top units, and the convergence history of the Arabian and Eurasian continental plates based on data established with the detrital zircon U-Pb and (U-Th)/(He-Pb) double dating methods. The targeted NW Zagros hinterland deposits are the Red Beds Series (potential equivalent of the Razak Fomration in Iran; Etemad-Saeed et al., 2020) that are situated below the Main Zagros fault, and the proto-Zagros Tanjero (Amiran), Kolosh (Amiran), and Gercus (Kashghan) Formations that occur in the Kurdistan region of Iraq (Fig. 1b). We will combine the new provenance information with published data from the wedge-top deposits of the Neogene Zagros basin including: the Fatha (Gachsaran), Injana (Aghajari), Mukdadiya (Lahbari Member-Lower Bakhtiari?), and Bai-Hasan (Bakhtiari) Formations, as well as the Lower Cretaceous Garagu Formation and the Paleozoic Khabour, Pirispiki Formations (Fig. 2). The ultimate goal is to track the provenance shifts in the erosional hinterland, which will build the bases to invoke the occurrence of specific geodynamic events throughout the Zagros orogeny including subduction, collision, and post-collisional uplift and exhumation.

# 2 Geologic background

## 2.1 The pre-Zagros orogeny strata

From the Cambrian and throughout most of the Paleozoic time (Fig. 2), the northeastern Arabian stratigraphic succession developed on the Pan-African basement and was part of the northern Gondwana realm along the Paleotethys

Ocean. The Paleozoic stratigraphic column contains mostly clastic sedimentary rocks with some marginal marine carbonate successions (Konert et al., 2001). Prior to the onset of the Hercynian orogeny during the late Devonian, the Arabian plate was considered as a stable platform. During the Hercynian orogeny, however, the platform experienced intracratonic deformation, which lead to several zones of subsidence and uplift (Sharland et al., 2001). During the Permo-Triassic time, the eastern margin of Gondwana experienced rifting and the Neotethys Ocean started to open. In the Kurdistan region of Iraq the exposed Paleozoic rocks include two main clastic formations, which are referred to as the Khabour (Ordovician) and Pirispiki (Late Devonian) Formations (Al-Hadidy, 2007). The Khabour Formation consists of thin-bedded, fine-grained sandstones, quartzites, and shales that were deposited in a shallow to deep marine environment (van Bellen et al., 1959; Al-Bassam, 2010). The Pirispiki Formation includes sandstones, siltstones, and red mudstone beds, as well as conglomerates, which have been interpreted as records of a partially marine to a fluvial environment, but mostly non-marine (van Bellen et al., 1959; Al-Hadide et al., 2002; Al-Hadidy, 2007; Al-Juboury et al., 2020).

After the opening of the Neotethys Ocean during the Permo-Triassic, the border of the Arabian plate developed into a passive margin that accumulated thick sequences of limestones, shales, and evaporite beds during most of the Mesozoic (Fig. 2) (Sharland et al., 2001; Stampfli and Borel, 2002; Ziegler, 2001; English et al., 2015). During the Middle Jurassic, the Gotnia basin was formed, which hosted deep-water carbonates and shales (Aqrawi et al., 2010). During the Late Jurassic, the west and northwestern part of Iraq experienced uplift, resulting in the formation of the Khlesia and Mosul basement highs, and the Gotnia basin configuration changed considerably (Ameen, 1992; Aqrawi et al., 2010; English et al., 2015). This variation in basin geometry resulted in the deposition of relatively thin (less than one hundred meters) carbonate rocks in association with clastic sandstone beds in the western side of the basin during the Early Cretaceous. Toward the eastern side of the basin, a basinal carbonate succession such as the Balambo Formation (~2000 m) was deposited (van Bellen et al., 1959; Jassim and Buday, 2006; English et al., 2015). Later during most of the Late Cretaceous, the basin was a site for further deposition and filled with shelfal and lagaoonal carbonates. During the Campanian-Maastrichtian time, the northern Arabian plate was affected by a rifting phase that lead to the formation of grabens filled with synrift marly deposits. In addition, an environment with shallow water carbonates established on structural highs (Aqrawi et al., 2010; English et al., 2015).

## 2.2 The proto-Zagros foreland basin strata

By the Maastrichtian time the proto-Zagros flexural foreland basin started to form in response to the arrival of the Neotethys intraoceanic subduction zone at the Arabia plate margin, leading to the ophiolite obduction (Stoneley, 1975; Koop and Stoneley, 1982; Alavi, 1994; Homke et al., 2010; Saura et al., 2011; Barber et al., 2019). This process resulted in a basin geometry where the sedimentary infill thinned toward the West with a surge of clastic input sourced mostly from the NW and the NE (Karim and Surdashy, 2005; Aqrawi et al., 2010; Çelik and Salih, 2021). The main clastic formations of the proto-Zagros foreland basin in the Kurdistan region of Iraq are the Tanjero, Kolosh, and Gercus Formations with the ages of Maastrichtian, Paleocene, and Eocene respectively (Fig. 2) (van Bellen et al., 1959; Abdel-Kireem, 1986). The reported nature of the contact between these formations is both conformable and unconformable (van Bellen et al., 1959; Jassim and Sisakian, 1978; Tamar-Agha et al., 1978; Al-Shaibani and Al-Qayim, 1990; Jassim and Buday, 2006; Abdallah and Al-Dulaimi, 2019; Kharajiany et al., 2019). The depositional environment of the proto-Zagros foreland basin strata was interpreted as distal to proximal deep-marine deposits that shallow upward to non-marine deposits (Al-Qayim, 1994; Al-Qayim et al., 2008; Jassim and Buday, 2006; Kadem, 2006; Al-Mashaikie et al., 2014). By the late Eocene, this clastic succession of the proto-Zagros foreland basin was covered by the shallow lagoonal marine carbonates of the Pila-Spi (Shahbazan?) Formation (van Bellen et al., 1959; Aqrawi et al., 2010). On top of the Pila-Spi Formation, an unconformity has been recorded based on absence of the Oligocene-early Miocene rocks that are linked to the collision between Arabia and Eurasia (Fig. 2) (Dunnington, 1958; Ameen, 2009; Lawa et al., 2013).

## 2.3 The Neogene Zagros foreland basin strata

During the Neogene, the continent-continent collision between the Arabian and Eurasian plates became prominent and consequently new clastic material was supplied into the new flexural basin (Koshnaw et al., 2017; 2020a). The synorogenic Neogene Zagros foreland basin formations are the Fatha (Gachsaran), Injana (Aghajari), Mukdadiya (Lahbari Member-Lower Bakhtiari?), and Bai-Hasan (Bakhtiari) Formations (Fig. 2). From bottom to top, the stratigraphic succession changes from the mixed clastic-carbonate-evaporite deposits of the Fatha Formation to the meandering and braided fluvial,

and alluvial deposits of the Injana, Mukdadiya, and Bai-Hasan Formations (Shawkat and Tucker, 1978; Tamar-Agha and Al-Aslami, 2015; Tamar-Agha and Salman, 2015; Koshnaw et al., 2020a).

## 2.4 The Red Beds Series strata

         The Red Beds Series (RBS) is located along the suture zone in the hinterland of the NW Zagros fold-thrust belt on the Arabian plate and overthrust by the Main Zagros fault (Fig. 1b). The RBS deposits consist of several NW-SE oriented

discrete outcrops where alternating mudstones, sandstones, conglomerate beds, as well as carbonate beds are encountered. They have been grouped into three major units: Suwais Group, Govanda Formation, and Merga Group. The Suwais and Merga Groups have been further subdivided into four and two subgroups, respectively, based on their lithofacies. Generally, the thickness of the clastic Suwais Group varies geographically between 1000 - 1500 m (Jassim et al., 2006). In some locations, this unit is even thicker. Overall, the Suwais Group represents a succession of dark red mudstones, siltstones, and

sandstones with frequently occurring conglomeratic beds. Generally, the suite coarsens and thickens upward so that the conglomerate beds with cobbles and boulders become the dominant lithofacies toward the upper part. The Suwais Group is regarded as marine to mostly non-marine fluvial deposits (Jassim et al., 2006; Alsultan and Gayara, 2016). A Paleocene-Eocene depositional age has been inferred for this unit based on fossil contents, for which, however, no list has been reported (Jassim et al., 2006), Recently, using detrital zircon ages as criteria, Koshnaw et al., (2019) inferred a maximum depositional

age as old as the late Oligocene (~26 Ma).

         The Suwais Group conformably overlies the Maastrichtian Tanjero Formation, but it was considered to follow the Lower Cretaceous Qulqula Formation with an unconformity (Karim et al., 2011; Hassan et al., 2014). However, using the late Oligocene maximum depositional age of the Suwais Group as criteria, Koshnaw et al. (2019) interpreted the contact with the Tanjero Formation as unconformable. The upper contact of the Suwais Group with the Govanda Formation is

reported as unconformable (Jassim et al., 2006). The Govanda Formation comprises ~100-120 m-thick reefal limestones with a few sandstone and conglomerate interebeds. Based on fossiliferous records, a Burdigalian to Langhian depositional age was proposed (van Bellen et al., 1959; Abdula et al., 2018). The ~200-500 m-thick Merga unit, which overlies the Govanda Formation (Fig. 2), consists of red mudstones and sandstones, some conglomerate interbeds with cobbles and boulders particularly in the upper ~200 m of the Merga Group. This group is considered to have been deposited in a fluvial

to alluvial environment (Jassim et al., 2006; Alsultan and Gayara, 2016). The contact between the Govanda Formation and Merga Group is conformable (van Bellen et al., 1959; Jassim et al., 2006).

         Along-strike of the Zagros suture zone toward Iran, the potential equivalent of the RBS, the Razak Formation, appears not to be preserved in the Lurestan segment, but in the Dezful and Fars segments it has been documented (James and Wynd, 1965; Alavi, 2004; Khadivi et al., 2010, 2012; Vergés et al., 2018; Etemad-Saeed et al., 2020). The Razak Formation

shows a lithostratigraphy and a timing of deposition that are similar to the RBS particularly in the hinterland. The stratigraphy includes carbonates and calcareous argillites toward the foreland, and shales, siltstones, sandstones and conglomerates toward the suture zone (Alavi, 2004). Available magnetostratigraphic chronologies, yet between the High Zagros fault and the Mountain Front Flexure and not adjacent to the Main Zagros fault, suggest 19.7 – 16.6 Ma as the time of deposition of the Razak Formation (Khadivi et al., 2010). The older deposits of the Razak Formation closer to the Main

Zagros fault might have been eroded due to deformation and exhumation (Alavi, 2004; Khadivi et al., 2012). These

characteristics regarding the lithology and the timing of deposition suggest that the Razak Formation could correspond to the Govanda Formation and the Merga Group of the RBS. However, unlike the RBS in the Kurdistan region of Iraq, the Razak Formation seems to be more wide spread geographically from the hinterland toward the foreland, and it appears in direct contact with the Asamri (Jeribe), Gachsaran (Fatha), and Aghajari (Injana) Formations (Alavi, 1994; Khadivi, 2010; Etemad-Saeed et al., 2020).

## 3 Structural and stratigraphic relationships with the Red Beds Series

In the NW Zagros suture zone, the Red Beds Series (RBS) are overthrusted by the Qandeel Series, the Walash Group, and the Naopurdan Group (Fig. 3). Along strike, these units are interchangeably thrust onto the RBS and they mark the Main Zagros fault (Fig. 1b). In the vicinity of the Main Zagros fault along the Tanun anticline (Figs. 4 and 5), angular unconformities are preserved between the older Upper Cretaceous formations and (i) the middle Miocene Govanda Formation on the northwestern plunge (Fig. 5b), and (ii) the upper Oligocene – lower Miocene (?) Suwais Group on the southeastern plunge (Fig. 5c). Along strike farther to the southeast, an angular unconformity between the Red Beds Series (Suwais Group) and the Lower Cretaceous Qulqula radiolarites is also preserved (Karim et al., 2011). Structurally, the highest unit is the Qandeel Series, which is an ophiolite-bearing Upper Cretaceous metamorphosed klippe that likely disjointed by an out-of-sequence deformation from the adjacent terranes of the Sanandaj-Sirjan zone (Jassim et al., 2006; Ali et al., 2014; Ali et al., 2019). The Walash Group (Figs. 3e and 6a,b), situated below the Qandeel Series thrust sheet (Fig. 3 and 4), is a volcaniclastic unit that is made up of dark red mudstones, shales, slates, relatively limited beds of sandstones and conglomerates, radiolarites and limestone beds, and metasediments near the top. These beds are associated with pyroclasts, basic pillow lavas (diabase rocks, pyroxene-bearing spilitic basalts, and spilites) and basaltic andesite flows (pyroxene, andesite and amphibole andesites) (Jassim et al., 2006; Ali et al., 2013). Pyroclastic feldspar grains of the Walash Group have been dated by the $^{40}Ar/^{36}Ar$ method and yielded $43.01 \pm 0.15$ Ma (Ali et al., 2013). Additionally, the $^{40}Ar/^{36}Ar$ age of the andesite lava flow is $43.1 \pm 0.3$ Ma, and the diabase rock is $40.1 \pm 0.3 - 32.3 \pm 0.4$ Ma (Aswad et al., 2014). The isochronal age of $^{87}Rb/^{86}Sr$ yielded ~44 Ma (Koyi, 2009). Depending on the planktonic and benthic foraminiferal assemblages, a marine environment with a depositional age of middle Eocene (Lutetian) has been proposed (Al-Banna and Al-Mutwali, 2008). The Walash Group itself was thrusted onto the Naopurdan Group, which is mostly made up of metasedimentary rock, but includes basalt flows and pillow lavas as well. The Naopurdan Group (Figs. 3e and 6c,d,e) is composed of sandstones, shales, and conglomerates with basic volcanic, nummulitic limestone, and slate constituents. The upper part consists of an upward coarsening succession from sandstones to conglomerates with basic volcanic clasts (Jassim et al., 2006; Ali et al., 2013). Based on the planktonic foraminiferal assemblage, a marine environment of Eocene age has been suggested (Jassim et al., 2006), but $^{40}Ar/^{36}Ar$ ages measured on basaltic feldspars yielded younger ages between $24.31 \pm 0.60$ Ma and $33.42 \pm 0.44$ Ma and (Ali et al., 2013). The Walash and Naopurdan Groups were correlated with similar rock units in the adjacent part of the Zagros belt in Iran and named as Walsh-Naopurdan-Kamyaran (WNK) (Ali et al., 2014; Moghadam et al., 2020). In the study area, the Naopurdan Group was thrust onto the autochthonous Red Beds Series deposits (Figs. 3, 4 and 6) on the Arabian plate. Along the thrust, the Naopurdan Group shows clear evidence for tectonic deformation such as slate pencil cleavage (Figs. 3e and 6e). At a different locality below the Naopurdan thrust sheet, the Merga Group shows a well-developed shear zone with c- and s-planes (Figs. 3e and 6f).

**4 Detrital zircon geochronology and thermochronology**

**4.1 Sampling and methods**

In this paper 1097 new detrital zircon U-Pb ages are presented from 11 samples (Supplemental tables 1 and 2),
which include eight samples from the Red Beds Series and three samples from the proto-Zagros formations. These new data
are integrated with previously published U-Pb ages in the study area (Koshnaw et al., 2019; Koshnaw et al., 2020a).
Additionally, for the purpose of the zircon (U-Th)/(He-Pb) double dating, 74 detrital zircons were selected for conducting
new (U-Th)/He analyses from these geochronologically dated grains. These minerals were extracted from five Red Beds
Series samples (Supplemental tables 1 and 3).

All sample preparation and analyses on the new samples were conducted at the University of Texas at Austin Geo-
and Thermochronometry and mineral separation laboratories. Mineral separation was conducted following mineral
separation procedure (e.g., Gehrels, 2000) and ~120 zircon grains of different size (30–100 μm) and shape were selected
randomly from each sample for further analysis. The zircon U-Pb ages were produced using the Laser Ablation Inductively
Coupled Plasma Mass Spectrometry (LA-ICP-MS) following procedures outlined in Marsh and Stockli (2015) and Hart et
al. (2016). In order to preserve the zircon grains for future (U-Th)/He analysis, the detrital zircon geochronologic ages were
obtained by depth-profile U-Pb analysis on unpolished grains (Stockli, 2017). The reported zircon U-Pb ages in this study
represent discordance ages <20 % with 2σ analytical uncertainty. The age concordance was determined based on the age of
$^{206}Pb/^{238}U$ for zircon grains <950 Ma, and the $^{207}Pb/^{206}Pb$ age for zircon grains >950 Ma. After completion of the U-Pb
analytical process, 13-21 representative zircon grains (except sample 12KRD-132, n=7) from major age peaks were
manually picked from the tape mount and packed in a platinum (Pt) pocket for zircon (U-Th)/He thermochronometric
analysis. In this study, the presented zircon helium ages are alpha ejection corrected (Farley et al., 1996) with standard
procedure uncertainty of ~8% and 2σ (Reiners et al., 2002). The zircon (U-Th)/He ages were acquired following the
methodology described in Wolfe and Stockli (2010). For detailed laboratory and data reduction processes please refer to
Thomson et al. (2017), Xu et al. (2017), and Pujols et al. (2020).

**4.2 Detrital zircon U-Pb geochronology results**

The detrital zircon U-Pb age distributions from three new proto-Zagros foreland basin samples manifest a unimodal
U-Pb age spectra of Early-Late Cretaceous time (~120-85 Ma) throughout the Tanjero (12KRD-142: ~120-90), Kolosh
(12KRD-143: ~105-85), and Gercus (12KRD-146: ~110-85) Formations, with an age peak of ~100 Ma (Fig. 7b). Among
these, the Kolosh sample 12KRD-143 contains two relatively young grains of middle Eocene (~40 Ma). Interestingly, the
detrital zircon U-Pb age populations that are determined for the Upper Cretaceous-Eocene proto-Zagros foreland basin fill
are very different compared to the older Paleozoic and Lower Cretaceous pre-Zagros orogeny deposits and the younger
Neogene Zagros foreland fill (Fig. 7a,c) (Koshnaw et al., 2020a). The detrital zircon U-Pb age results from the eight new
Red Beds Series samples display a bimodal signature with late Eocene (~35-45 Ma) and Paleocene (~55-65 Ma) ages with
peaks of ~40 Ma and ~60 Ma, respectively (Fig. 8). Less commonly, an older age peak of ~100 Ma is also found, except a
samples from the Suwais Group (SH17S2) that shows a dominant ~100 Ma peak.

**4.3 Detrital zircon (U-Th)/He thermochronology and (U-Th)/(He-Pb) double dating results**

To further constrain the potential source regions for the Red Beds Series deposits based on the zircon (U-Th)/(He-
Pb) double dating, five samples were chosen for the zircon (U-Th)/He thermochronometric analysis (ZHe). Two samples
were collected from the NW of the study area and three samples from the SE of the study area, including the Suwais and
Merga Groups in each area (Fig. 9). Detrital zircon grains were selected from each main U-Pb age peaks including: ~40 Ma

(late Eocene), ~60 Ma (middle Paleocene), ~75 Ma (Late Cretaceous), ~100 Ma (late Early Cretaceous), ~170 Ma (Jurassic), ~300 Ma (Carboniferous-Permian), and ~600 Ma (Neoproterozoic). Generally, the bulk ZHe age results display notable ranges of ~66-55 Ma with an age peak of ~60 Ma, and of ~50-35 Ma with age peaks of ~40 Ma for the Suwais Group, and ~45 and ~37 Ma for the Merga Group (Fig. 9a). Two grains from the Merga Group that are extracted from the samples MT17M3 and SH17M5 chronicle the youngest exhumation at ~21 Ma. From the Suwais Group, three ZHe ages that overlap within error mark the youngest peak of ~29 Ma. The detrital zircon (U-Th)/(He-Pb) double dating results reveal cooling ages of ~40-30 Ma for the late Eocene grains and ~60-40 Ma for the middle Paleocene grains (Fig. 9b,c). The Late and late Early Cretaceous zircon grains have cooling ages of ~70-35 Ma. The Jurassic zircon grains show cooling ages of ~60-40 Ma, which are comparable to the middle Paleocene zircon grains. The Neoproterozoic and Carboniferous-Permian zircon grains manifest a wide range of cooling ages of ~150-30 Ma and ~120-50 Ma, respectively.

## 5 DISCUSSION

### 5.1 Sedimentary provenance evolution

#### 5.1.1 Provenance of the NW Zagros wedge-top deposits

The new detrital zircon U-Pb age results form this study highlight distinctive signatures for each of the various tectonic phases including: (i) the passive margin of the pre-Zagros orogeny, the proto-Zagros foreland basin stage related to the obduction of the Neotethys oceanic crust, and (iii) the Neogene Zagros basins related to the Arabia-Eurasia continental collision (Fig. 10a,b). In particular, the provenance data from the Maastrichtian Tanjero (Amiran), the Paleocene-Eocene Kolosh (Amiran), and the Eocene Gercus (Kashghan) Formations, all of which were deposited in the proto-Zagros foreland basin, show a strong influence of Neotethys-related terranes and a striking difference with the older and younger formations, implying a switch of the main sediment sources through time. The key source terranes in the NW Zagros and the surrounding areas are (i) the Paleocene-Eocene Walash-Naopurdan-Kamyaran complex (WNK), (ii) the Cretaceous Neotethys oceanic crust and the island arc ophiolitic terranes, (iii) the Jurassic and Triassic igneous rocks within the Sanandaj-Sirjan zone (SSZ), (iv) the Carboniferous-Permian northern Gondwana and Variscan rocks, and (v) the Neoproterozoic Pan-African/Arabian-Nubian shield and older rocks (Fig. 7).

Samples from the proto-Zagros foreland basin formations that were obtained from the NW of the study area (Fig. 1b) have a sole age peak of ~100 Ma. This age peak points to the Neotethys oceanic crust and island arc ophiolitic terranes as principal sources during the Maastrichtian and the Paleogene times. The pre-Zagros passive margin strata (Fig. 7) from the northern part of the study area have detrital zircon U-Pb age distributions dominated by the early Neoproterozoic, Pan-African, Cadomian (peri-Gondwana), and other Gondwana-related sources (Stern and Johnson, 2010; Alirezaei and Hassanzadeh, 2012; Stern et al., 2014; Avigad et al., 2016, 2017; Golan et al., 2017; Koshnaw et al., 2017; Al-Juboury et al., 2020; Omer et al., 2021). These age components are also found in the detrital zircon populations of the Lower Cretaceous formations in the southern Iraq (Wells et al., 2017). The late Paleozoic age components from the pre-Zagros formations are likely Gondwana-related, unlike the comparable age components from the younger formations that likely involve Variscan-derived detritus (e.g. Barber et al., 2019). In general, the sediment of the Paleozoic and Lower Cretaceous rocks were largely originating from (i) the Arabian-Nubian shield, (ii) the Gondwana super-fan system, which itself was sourced from the East African Orogen, or from (iii) the SE Saharan Metacraton (Koshnaw et al., 2017; Meinhold et al., 2021).

Overall the contrast in the detrital zircon provenance signature between the pre-Zagros and proto-Zagros deposits reflect a fundamental reversal in the sediment source from the West to the East as the basin started to flex toward the East. This flexural downbending most likely reflects the arrival of the Neotethys oceanic curst and ophiolitic terranes, which was associated with numerous Late Cretaceous magmatism, including granitoid intrusions with zircons of a middle Cretaceous age (Delaloye and Desmons, 1980; Ali et al., 2012; Nouri et al., 2016; Al Humadi et al., 2019; Ismail et al., 2020). In the SE

of the study area, detrital zircons from the Tanjero Formation with Paleozoic and Proterozoic ages, as well as few grains with Triassic and latest Paleocene ages have been reported (Jones et al., 2020). Farther SE, in Iran, the proto-Zagros strata of the Amiran and Kashghan Formations have ~240 Ma Triassic and middle Cretaceous ~100 Ma age signatures in their detrital records (Zhang et al., 2016; Barber et al., 2019). These additional old detrital zircon age components potentially imply a local variation of the paleogeography along-strike. Even though zircon is not a common mineral in mafic rocks, yet it has been recognized (Grimes et al., 2007 and references therein). The ~240 Ma Triassic age signature in the detrital zircon records of the Amiran and Kashghan Formations has been attributed to the mid-oceanic ridge based on the trace element data measured in zircon grains (Barber et al., 2019). Additionally, sandstone petrographic analysis on the Tanjero Formation revealed the occurrence of limestone and chert grains that were particularly sourced from the older Qulqula radiolarite and Balambo Formations in the NW (Aziz and Sadiq, 2020; Jones et al., 2020; Çelik and Salih, 2021). Such variation in the sediment source for the proto-Zagros foreland basin is in line with (i) the destruction of the Gotnia basin architecture (ii) an input of carbonate material into the new flexural basin, and (iii) the occurrence of some recycled Paleozoic and older zircon grains in the Tanjero Formation (Aqrawi et al., 2010; English et al., 2015; Aziz and Sadiq, 2020; Jones et al., 2020). The younger Neogene Zagros foreland basin strata have also distinct U-Pb age spectra that are dissimilar to the proto-Zagros and the pre-Zagros U-Pb age distributions. In the same locality where the new proto-Zagros samples were collected (Fig. 1b), published data show that most of the Neogene formations, except the Injana Formation, are characterized by an Eocene (~40 Ma) age peak with subordinate contributions of older ages to the age spectra (Fig. 7). However, in different localities, the Neogene deposits show a more mixed signature (Koshnaw et al., 2020a). These sediments with zircon grains recording a dominant Eocene (~40 Ma) age peak were interpreted to have been supplied by transverse rivers from the NE during a late stage of the Zagros orogeny (Koshnaw et al., 2020a). In addition to the Eocene age peak, the Injana Formation records a wider range of source terranes where the detrital zircons are characterized with younger early Oligocene ages and older age components. This suggests that the sedimentary material was likely recycled from older uplifted strata and delivered by axial rivers from the NW (Koshnaw et al., 2020a).

Ultimately, such variation in the detrital zircon U-Pb provenance between the different hinterland wedge-top units of the NW Zagros basin denote that (i) the pre-Zagros passive margin deposits are diagnosed by a wide age range of old zircon grains, (ii) the proto-Zagros sediments that are related to the Neotethyan obduction record a unique zircon age component of Albian-Cenomanian, and that (iii) the Neogene Zagros foreland basin deposits are dominated by zircon grains with Eocene and early Oligocene ages.

**5.1.2 Provenance of the NW Zagros Red Beds Series deposits**

The Red Beds Series (RBS) provenance data put an age limit on the diminution of the magmatic activities and on the exhumation of the source terranes. The detrital zircon U-Pb age populations from the RBS depict two characteristic age components, which are Paleocene ages with a peak of ~60 Ma and late Eocene ages with a peak of ~40 Ma (Fig. 8). Additionally, older subordinate age peaks similar to the pre- and proto-Zagros age components are also present. Zircon grains from the Paleocene and Eocene age components record identical (U-Th)/He and U-Pb ages, pointing to a volcanic origin (Fig. 9b,c). Furthermore, their cooling over ~60-40 Ma for the middle Paleocene grains and ~40-30 Ma for the late Eocene grains suggest the occurrence of a systematic exhumation and unroofing of magmatic plutons through time.

The older subordinate age components possibly reflect the combined effect of recycling of the previously deposited strata as shortening continued and local variations of rock outcrops along-strike, including potential exposure of the ophiolitic rocks. The potential source terrane candidates that have ages comparable to the Paleocene and Eocene age components are the Walash-Naopurdan volcaniclastic complex and Urumieh-Dokhtar magmatic zone (UDMZ), which both are located to the NE of the study area. The Paleogene Walash-Naopurdan volcaniclastic complex extends along-strike into Iran and was previously termed as the Walash-Naopurdan-Kamyaran (WNK) (Ali et al., 2014; Moghadam et al., 2020). The

WNK complex potentially extends into Turkey as well and is known as the Maden-Hakkari complex (Braud and Ricou, 1975; Yılmaz, 1993; Robertson et al., 2007; Oberhänsli et al., 2010). In Iran, the Walash-Naopurdan complex is equivalent to the volcaniclastic deposits near Kamyaran and Kermanshah, which have been named differently as Gaveh-Rud domain (Braud and Ricou, 1975; Homke et al., 2010; Saura et al., 2015), Early Tertiary magmatic domain (Agard et al., 2005, 2011), and Kamyaran Paleocene-Eocene arc (Whitechurch et al., 2013). These Paleogene rocks represent arc-related (defined differently as arc, backarc, and forearc) volcanic and volcaniclastic units (Homke et al., 2010; Ali et al., 2013; Whitechurch et al., 2013; Saura et al., 2015), which likely developed in the proximity of the southwestern margin of the Eurasian plate in association with the subduction of the Neotethys plate beneath Eurasia (Robertson et al., 2007; Barrier and Vrielynck, 2008; Homke et al., 2010; Agard et al., 2011; Whitechurch et al., 2013; Saura et al., 2015). The dated magmatic bedrocks within the WNK complex and part of the adjacent Sanandaj-Sirjan-Zone (SSZ) show both age components that are found in the RBS, i.e., Paleocene (Braud, 1987 in Homke et al., 2010; Azizi et al., 2011; Whitechurch et al., 2013) and Eocene (Mazhari et al., 2009, 2020; Azizi et al., 2011, 2019; Ali et al., 2013; Aswad et al., 2014; Moghadam et al., 2020). The Eocene intrusions are also found within the Upper Cretaceous ophiolitic rocks (Leterrier, 1985; Aswad et al., 2016; Ali et al., 2017; Ismail et al., 2020). Younger Oligocene magmatic rocks within the WNK complex have also been reported (Whitechurch et al., 2013; Ali et al., 2013). The UDMZ contains large Eocene and Oligocene igneous bodies marking a prolonged magmatic flare-up (Verdel et al., 2011; Chiu et al., 2013) with a peak magmatism age of ~40 Ma (van der Boon et al., 2021). Accordingly, both the WNK complex and the UDMZ could have served as source areas for the Red Beds Series. However, the WNK complex was thrusted onto the RBS and it is adjacent to the SSZ, whereas the SSZ is situated between the RBS and the WNK complex to the West and the UDMZ to the East, which makes any sedimentary pathways from the UDMZ more complicated. Additionally, the regional uplift of the SSZ occurred earlier than that of the UDMZ, it makes it very unlikely that large volumes of UDMZ material were supplied to the depocenter of the Red Beds Series since uplift of the SSZ disconnected the sedimentary pathways (François et al., 2014; Barber et al., 2018). Moreover, the majority of the double-dated detrital zircons from the RBS that have Ediacaran, Carboniferous-Permian, and Jurassic ages show an exhumation age that corresponds to the time span between the Late Cretaceous and the Eocene (Fig. 9b), which is comparable to the exhumation age of the Sanandaj-Sirjan zone (e.g. Homke et al., 2010; Khadivi et al., 2012; Mouthereau et al., 2012; Barber et al., 2018). Such exhumation pattern of the old zircon grains that are found in the WNK-derived RBS supports the interpretation that the development of the WNK complex occurred in the vicinity of the Sanandaj-Sirjan zone.

The new data presented in this paper in combination with the previous data by Koshnaw et al. (2019) of the RBS illustrate (Fig. 8) that the youngest zircon age peak component of the Suwais Group samples is between ~37 Ma and ~41 Ma. In the same sense, the youngest zircon age peak component of the Merga Group samples is between ~36 Ma and ~45 Ma, however a younger ~33 Ma peak (three grains) also exists. Along-strike, both groups show older ages toward the SE, similar to the age trend of the Walash and Naopurdan rocks, ~24-33 Ma in the NW and ~32-43 Ma in the SE (Ali et al., 2019). Published detrital zircon U-Pb data (Jones et al., 2020) of three samples collected from the Tanjero Formation (T, n = 28), Suwais Group (LR, n = 5), and Merga Group (UR, n = 39) but farther to the SE of our study area show disparity with the provenance results from this study. This mainly concerns the samples from the Tanjero Formation and Suwais Group as they lack the middle Cretaceous and the Paleogene age components, respectively. This variation is likely due to the limited number of the analyzed grains, as dating less than 117 detrital zircon grains for provenance studies increases the chance of not detecting age components that consist of more than 5% of the population at 95% confidence level (Vermeesch, 2004).

So far in the vicinity of the study area, the late Oligocene magmatism has been documented only in two localities in the NW Zagros, including the Kurdistan region of Iraq and Iran (Ali et al., 2013; Whitechurch et al., 2013), but in Iran and thus farther NE of the study area, magmatism is more abundant (Agard et al., 2011; Chiu et al., 2013). In general, magmatic activity appears to be considerably less during the late Oligocene than during the Eocene (van deer Boon et al., 2021). In the Kurdistan region of Iraq, earlier partial cessation of magmatism in the southeastern part is potentially due to the presence of

the Bisotoun block (Agard et al, 2005, Wrobel-Daveau et al., 2010; Sasvari et al., 2015), which implies that the thrusting of the WNK terranes occurred earlier farther SE of our study area. As the geographic extension of the Bisotoun block is limited, this local difference in the timing of the WNK terrane thrusting can not be in contradiction with the regional oblique Arabia-Eurasia collision that potentially commenced first farther in the NW, away from the study area (e.g. Rolland et al., 2011; Darin et al., 2018).

Overall, the U-Pb age populations of the Red Beds Series deposits show consistent Paleocene and Eocene age signatures from the bottom to the top, which imply that the WNK complex was a major source area and that the sediment sources have not changed significantly during deposition of the RBS.

## 5.2 Basin dynamic of the NW Zagros Red Beds Series deposits

The angular unconformity between the Suwais Group and the Upper Cretaceous strata, and the detrital zircon U-Pb and (U-Th)/(He-Pb) double dating results advocate for deposition of the RBS in a distinctive hinterland basin.

On the basis of the documented growth strata in the Masstrichtian Tanjero Formation of the Tanun anticline (Le Garzic et al., 2019) and the lack of growth strata in the Suwais Group (Figs. 4 and 5c), the Suwais Group deposition must have been taken place after the deposition of the Tanjero Formation and the initial growth of the Tanun anticline. This period of no stratigraphic record is also evident from the ~26 Ma detrital zircon maximum depositional age of the Suwais Group (Koshnaw et al., 2019) and the Masstrichtian age of the Tanjero Formation (Abdel-Kireem, 1986). However, no direct angularity is manifested between the Tanjero Formation and the Suwais Group (Le Garzic et al., 2019), possibly because of erosion of the tilted beds of the Tanjero Formation where the Suwais Group is deposited and the preservation of the semi-horizontal beds. The angular unconformity between the Suwais Group and the Upper Cretaceous beds (Fig. 5c) along with other angular unconformities between the Suwais Group and the Qulqula Formation (Karim et al., 2011), and the Govanda Formation and the Upper Cretaceous beds (Fig. 5b) suggest that the RBS unit was deposited in an intermontane setting after the development of the proto-Zagros fold-thrust belt (Lawa et al., 2013; Koshnaw et al., 2017, 2020b; Le Garzic et al., 2019).

In this study, the comparison of the detrital zircon U-Pb provenance data established for the RBS, pre-Zagros, proto-Zagros, and Neogene Zagros sediments reveals a robust dissimilarity of the RBS deposits with the pre-Zagros and proto-Zagros units, but a possible linkage with the Neogene Zagros units (Fig. 10). On the basis of the sandstone petrography, the Suwais Group and the Tanjero Formation, are considered as litharenites, yet a key difference is the presence of the magmatic and metamorphic grains in the Suwais Group and their scarcity in the Tanjero Formation (Hassan et al., 2014; Çelik and Salih, 2018, 2021).

To further constrain the tectonic setting of the RBS, three principal hypotheses can be tested: (a) the RBS deposited in a foreland basin; (b) the RBS deposited in a foreland basin, but later developed into an intermontane basin; (c) and the RBS deposited in an intermontane basin (Fig. 11). To render a unique solution, the detrital zircon (U-Th)/(He-Pb) double dating method has been utilized. In scenario a (Fig. 11a), the RBS deposits are expected to have a similar signature as the proto-Zagros foreland basin where the detrital zircon U-Pb age spectra is dominated by a ~100 age peak (this study) and characterized by a ~70 Ma zircon (U-Th)/He age peak (Barber et al., 2019),. For scenario b (Fig. 11b) where the RBS deposition would take an early stage foreland and a late stage intermontane basin position, we would expect zircon grains with a mixed U-Pb age distribution, which should reflect a mixture of: (i) recycled grains with ~100 Ma and ~70 Ma zircon (U-Th)/He age peaks, and (ii) new ~60 Ma and ~40 Ma grains with their corresponding zircon (U-Th)/He ages. In scenario c (Fig. 11c) where the tectonic setting of the RBS is unrelated to the foreland basin, a mixed U-Pb age spectra similar to the scenario b is predicted, yet with no inherited ~70 Ma zircon (U-Th)/He age because no recycling of foreland basin strata is expected. The scenario a can be ruled out as the RBS detrital zircon U-Pb age spectra is substantially different (Figs. 7,8, and 10). Scenario b and c offer valid alternatives, but there are two key differences in the U-Pb and (U-Th)/He ages. In scenario

b, because the RBS basin involves an early stage foreland basin, no detrital zircons with Paleocene and Eocene ages should be expected in its lower strata. Yet samples from the Suwais Group are dominated by the Paleocene and Eocene age components (Fig. 8). Additionally, the detrital zircon (U-Th)/(He-Pb) double dating results (Fig. 9) show that most of the ~100 Ma U-Pb age grains have a ~65 Ma and ~40 Ma exhumation age that is comparable to the Paleocene and Eocene age grains. These ages are still younger than the ~70 Ma zircon (U-Th)/He age of the proto-Zagros strata. Moreover, due to presence of the ~40 Ma zircon (U-Th)/He age and insufficient burial in the proto-Zagros basin to reset the zircon (U-Th)/He ages, they are unlikely to record an origin from the deformed proto-Zagros strata. Indeed, the pre-deformational thickness of the proto-Zagros formations collectively (Tanjero, Kolosh, Gercus Formations) is about 3-3.6 km (van Bellen et al., 1959). Assuming $20^o$ for the surface temperature and $25^o$ C/km for the geothermal gradient, about 6-7 km thickness would be required to reset zircon the (U-Th)/He ages. Accordingly, the ~100 Ma U-Pb ages encountered in detrital zircon grains rather reflect the supply of material from the newly eroded ophiolitic terranes where igneous intrusions have a similar age. Therefore, on the basis of the detrital zircon U-Pb signature and the (U-Th)/(He-Pb) double dating results, we argue that scenario c, the intermontane basin hypothesis, is more plausible and compatible with the provenance characteristics of the RBS deposits.

All-inclusive, the angular unconformities and the provenance information suggest neither a significant connection of the RBS intermontane basin with the proto-Zagros foreland basin nor a potential connection during the Miocene with the Neogene Zagros foreland basin, possibly as a result of a broad Zagros collisional zone where uplift and exhumation occurred. This interpretation about the position of the RBS in the NW Zagros in the Kurdistan region of Iraq is most likely also valid for the Razak Formation in the Dezful and Fars areas of Iran, yet under the consideration of a possible variation in the structural architecture of the basin. In the study area, the RBS is structurally bound by the allochthonous WNK complex toward the NE and by the anticlines of the proto-Zagros fold-thrust belt toward the SW (e.g. Figs. 4 and 5), and the RBS deposits overlay the Paleogene and older rocks in direct contact. However in the Dezful and Fars segments of the Zagros inIran, where the development of the Paleogene proto-Zagros fold-thrust belt is limited (Hessami et al., 2001), the Razak Formation deposits occur in direct contact with the Neogene Zagros foreland basin deposits with no documented unconformities (James and Wynd, 1965; Khadivi et al., 2010; Vergés et al., 2018; Etemad-Saeed et al., 2020). Such a basin setting suggests that the Razak Formation could be interpreted as an early foreland basin unit at least during the Neogene. In either case, we emphasize that both the RBS and the Razak Formation, are representing the deposits of the Arabia-Eurasia early stage collision (Khadivi et al., 2012; Koshnaw et al., 2019; Etemad-Saeed et al., 2020).

**5.3 Geodynamic evolution**

The combination of new field evidence with the provenance data and with the existing knowledge about the magmatic and tectonic history of the Arabian and Eurasian margins allows us to unravel the Cenozoic convergence history between the two plates in the present-day NW Zagros (Figs. 12 and 13).

After the initiation of the Neotethys intraoceanic subduction zone during the Aptian, its development until the middle Late Cretaceous, including the growth of ophiolitic island arcs (Aswad and Elias, 1988; Aziz et al., 2011; Ali et al., 2012; Barber et al., 2019), the ophiolitic terranes obducted on the northeastern margin of the Arabian plate during the Campanian (Figs. 12 and 13a). Sediments were shed from the uplifted terranes on the newly formed proto-Zagros foreland basin as evidenced from the detrital zircon provenance data from this study and the detrital zircon (U-Th)/He ages from the Amiran and Kashghan Formations from Lurestan in Iran (Barber et al., 2019). In the Upper Cretaceous Bekhma Formation (late Campanian-early Maastrichtian age; van Bellen et al., 1959), the U-Pb dating of hydrothermal cementation yielded ~74 Ma (Salih et al., 2019), consistent with the detrital geochronological and thermochronological constraints (Fig. 12). These lines of evidence suggest that the proto-Zagros fold-thrust belt growth initiated as a result of the ophiolite obduction. During the Paleocene (Figs. 12 and 13a), the WNK complex possibly originated as a back-arc to arc system, associated with

magmatic activity (Braud, 1987 in Homke et al., 2010; Azizi et al., 2011, 2019; Ali et al., 2013; Whitechurch et al., 2013). The signature of the Paleocene magmatism is also reflected in the detrital zircon U-Pb and (U-Th)/He records of the Red Beds Series that derived through erosion from the WNK complex. The locality of this arc system is contested. In particular, it is not clear whether it developed in the vicinity of the Eurasian plate (Braud and Ricou, 1975; Agard et al., 2005; 2011;

Robertson et al., 2007; Barrier and Vrielynck, 2008; Oberhänsli et al., 2010; Al-Qayim et al., 2012; Whitechurch et al., 2013) or close to the Arabian plate (Aswad et al., 2014; Ali et al., 2017, 2019). However, due to presence of Eurasian derived detritus (Homke et al., 2010; Whitechurch et al., 2013) and the occurrence of low-grade metamorphic grains (Ali et al., 2017), which are abundant in the Sanandaj-Sirjan zone (SSZ) (Hassanzadeh and Vernicke, 2016), a Eurasian-related setting that resulted from the oceanic slab rollback is more likely (Robertson et al., 2007; Agard et al., 2011; Whitechurch et

al., 2013; Abdollahi et al., 2020).

During the Eocene (Fig. 13b), the magmatism in the WNK complex and the SSZ edge continued, however the SSZ magmatism was mostly associated with intrusions rather than volcanism (Moritz et al., 2006; Mazhari et al., 2009, 2020; Azizi et al., 2011, 2019; Ali et al., 2013; Whitechurch et al., 2013; Aswad et al., 2014; Abdollahi et al., 2020; Moghadam et al., 2020). This period of magmatism in the WNK complex was coeval with the nearby voluminous UDMZ magmatic flare-

up (Verdel et al., 2011; Chiu et al., 2013; van der Boon et al., 2021), implying a genetic relationship. The downgoing oceanic slab was possibly subducted deeper and farther beneath the Eurasian plate with a relatively shallower angle that led to the generation of magmas in the WNK and UDMZ at a comparable period (Whitechurch et al., 2013; Hassanzadeh and Vernicke, 2016). Additionally, in the proximity of the Arabian plate, the Cretaceous ophiolitic terranes also contain Eocene intrusions (Fig. 13b) (Leterrier, 1985; Aswad et al., 2016; Ali et al., 2017; Ismail et al., 2020), with an origin that is likely

different to that of the WNK complex (Aswad et al., 2016). These Eocene intrusions in the Cretaceous ophiolitic terranes are likely the consequence of the breakoff of the subducted oceanic slab. Tomographic data for the Middle East show the occurrence of cold material at deep levels that has been considered as a remnant from an older early Paleogene slab breakoff event (Agard et al., 2011; van der Meer et al., 2018). This inferred cold material is different from the remnant of a younger (Miocene) and shallower slab breakoff (Mesopotamia and Zagros slabs; van der Meer et al., 2018).

For this period, the Eocene, several models suggested multiple subduction zones with 200-300 km distance between the zones across the Neotethys to explain the documented magmatic activities (Aswad et al., 2016; Ali et al., 2017; Ali et al., 2019). However, due to a space problem and difficulties of subduction inception a multiple subduction system has been considered unlikely (Whitechurch et al., 2013). Present-day natural examples of one single complete suite including a trench, a forearc, an arc, and a back-arc necessitate a horizontal width of 200-500 km, depending on the subduction angle (e.g. Cloos

et al, 1993; Stern, 2002, 2010). In this study the plate kinematic reconstruction software Gplates 2.2.0 (Müller et al., 2018) has been utilized and the published plate circuit model of Arabia-Eurasia (McQuarrie and Hinsbergen, 2013) was used as a base for the reconstruction, with the assumption of ~35 Ma as an onset of collision. Based on this setting, the calculated distance between the Arabian and Eurasian margins was estimated to be 283 ± 66 km at ~40 Ma. For this calculation, the length of the Harsin (35 km) and Bisotun (49 km) domains were excluded (Vergés et al., 2011), as the study area is farther

NW of Lurestan and no evidence has been found regarding their occurrence. Therefore, we hypothesize that the 283 ± 66 km distance between the Arabian and Eurasian plates is more compatible with a convergence model that involves a subduction of the Neotethys slab below Eurasia, and older subducted slab below the obducted ophiolite on Arabia. This conclusion is in line with the plate kinematic reconstruction models in the broader Middle East (Dewey et al., 1973; Barrier and Vrielynck, 2008; Jagoutz et al., 2015; Hinsbergen et al., 2020).

The latest Eocene to the late Oligocene (~36-26 Ma) marks a period of significant reduction of magmatism in the UDMZ, SSZ, and the WNK complex, and in the Late Cretaceous ophiolitic terranes (Figs. 12 and 13c). During the same time, a widespread unconformity was recorded in the NW Zagros in the Kurdistan region of Iraq (Dunnington, 1958; Ameen, 2009; Lawa et al., 2013). Furthermore, the development of the syndepositional compressional joints within the

upper Eocene Pila-Spi Formation has also been documented (Numan et al., 1998) in the Zagros fold-thrust belt. Lastly,
fractures that were filled by hydrothermal cementation in the Zagros fold-thrust belt hint to a period of fluid flow during the
early Oligocene (~30 Ma) as constrained by the U-Pb method (Salih et al., 2019). Among the presented RBS samples, there
are four young detrital zircon age peaks of ~36-37 Ma (Fig. 8). Such a pattern in the detrital zircon ages signifies a
remarkable attenuation of magmatism in the source terranes by ~36 Ma. However, the occurrence of a single younger peak
of ~33 Ma, and limited younger magmatic activity of ~26 Ma (Whitechurch et al., 2013) and ~24 Ma (Ali et al., 2013) imply
that magmatic activities continued locally. Samples from the RBS, particularly from the bottom section of the Suwais Group
in the study area (e.g. Fig. 5c) include the Eurasian detrital zircon age signature and contain zircon grains as young as ~26
Ma (Koshnaw et al., 2019). We argue that these data sets constraints on the timing of the Arabia-Eurasia continental
collision between ~36 Ma and ~26 Ma, which includes the entire geodynamic development from the commencement to the
termination in the NW Zagros, a conclusion which is in agreement with previous studies (Hempton, 1987; Agard et al.,
2005, 2011; Horton et al., 2008; Mark and Armstrong, 2008; Ballato et al., 2011; Mouthereau et al., 2012; Mohammad et al.,
2014; Barber et al., 2018).

After the continental collision, the Zagros suture, as well as the SSZ and the UDMZ underwent notable uplift and
exhumation starting the earliest Miocene (Figs. 12 and 13d). The youngest detrital zircon grains from the RBS deposits
(Merga Group) record a ~21 Ma zircon (U-Th)/He age, comparable to (i) the ~23 Ma zircon (U-Th)/He age of an Eocene
intrusion within the Cretaceous ophiolite (Ismail et al., 2020), (ii) the period of accelerated exhumation in the SSZ and
UDMZ as documented by apatite (U-Th)/He (AHe), apatite fission track (AFT), and zircon fission track (ZFT) ages
(François et al., 2014; Behyari et al., 2017; Barber et al., 2018), and (iii) a ~22 Ma AFT detrital age from the Aghajari
Formation of the Neogene Zagros foreland basin in Lurestan in Iran (Homke et al., 2010). Additionally, the earliest Miocene
is also the time of alkaline magmatism initiation in the UDMZ (Homke et al., 2010 and references therein; Mouthereau et al.,
2012 and references therein; van der Boon, 2021). Furthermore, the present-day field relationship reveal that in some
localities the Cretaceous ophiolitic terranes, including the associated Eocene intrusions that have a different petrochemistry
than the Eocene WNK, were thrusted onto the WNK (e.g. Figs. 3 and 4) likely by an out-of-sequence mechanism (Ali et al.,
2014; Aswad et al., 2016). Such an event likely occurred at a similar time to the documented ~21-23 Ma exhumation age
across the NW Zagros collisional zone as a consequence of the terminal collision. Moreover, in the southeastern Fars
segment of the Zagros belt, the AFT detrital age form the potential stratigraphic equivalent of the RBS, the Razak
Formation, yielded ~25 Ma, and it has been interpreted to record the exhumation of the Main Zagros hanging wall in the
suture zone (Khadivi, et al., 2012). Such a similarity in the suture zone exhumation during the latest Oligocene and the
earliest Miocene across the Zagros ~2000 km-long orogen may suggest a uniform suturing between Arabia and Eurasia, and
possibly reduction of the oblique collision effect and onset of the northward motion of Arabia (McQuarrie et al., 2003;
Navabpour et al., 2013). Later, during the middle and late Miocene (~14-4 Ma) further enhanced exhumation took place
from the suture zone forelandward with sediment inputs from the NW and the NE of the study area, which possibly triggered
by a new slab breakoff event (Koshnaw et al., 2020a,b).

**5 Conclusions**

An angular unconformity between the Red Beds Series (RBS) units, the Suwais Group and the Govanda Formation,
and the deformed Upper Cretaceous formations in the NW Zagros hinterland denotes deposition of the RBS in an
intermontane basin after the development of the proto-Zagros fold-thrust belt. The detrital zircon U-Pb age data of the RBS
units (Suwais and Merga Groups) and the proto-Zagros foreland basin strata (Tanjero, Kolosh, and Gercus Formations) are
strikingly different, yet the RBS data are partially commensurate with the Neogene Zagros foreland basin fill, and all data
sets are unrelated to the pre-Zagros Paleozoic and Lower Cretaceous record. The detrital zircon (U-Th)/(He-Pb) double

dating result pattern of the RBS further highlight that these sediments were deposited an intermontane basin. The dating results additionally demonstrate a signal related to the supply of volcanic material and to the unroofing of potential plutons, dissimilar to the proto-Zagros detrital record. The evaluation of the provenance data shows a derivation of the RBS deposits form the Paleogene Walash-Naopurdan-Kamyaran complex that was developed along the Eurasian margin, whereas the proto-Zagros deposits were derived from the Upper Cretaceous ophiolitic terranes. Significant cutback of magmatism by ~36 Ma, as reflected in the RBS youngest U-Pb age peaks, points to the latest Eocene (<36 Ma) as the onset of the Arabia-Eurasia continental collision in the NW Zagros. Altogether, the provenance shifts recorded in the pre-Zagros, proto-Zagros, and Neogene Zagros units, and in the WNK-derived RBS intermontane basin deposits reflect different phases of the Cenozoic Zagros orogeny.

**Acknowledgments**

This research was partially funded by the State Secretariat for Education, Research and Innovation of Switzerland via the Swiss Government Excellence Scholarship awarded to R. Koshnaw. We would like to thank I. Ahmed, V. Sissakian, M. Tamar-Agha, W. Shingaly, N. Karo, M. Zebari for discussion and logistical support. We extend gratitude to R. Chatterjee, D. Patterson, and L. Stockli at the University of Texas at Austin UTChron laboratories for their assistance. We thank Frédéric Mouthereau and an anonymous reviewer for their insightful reviews.

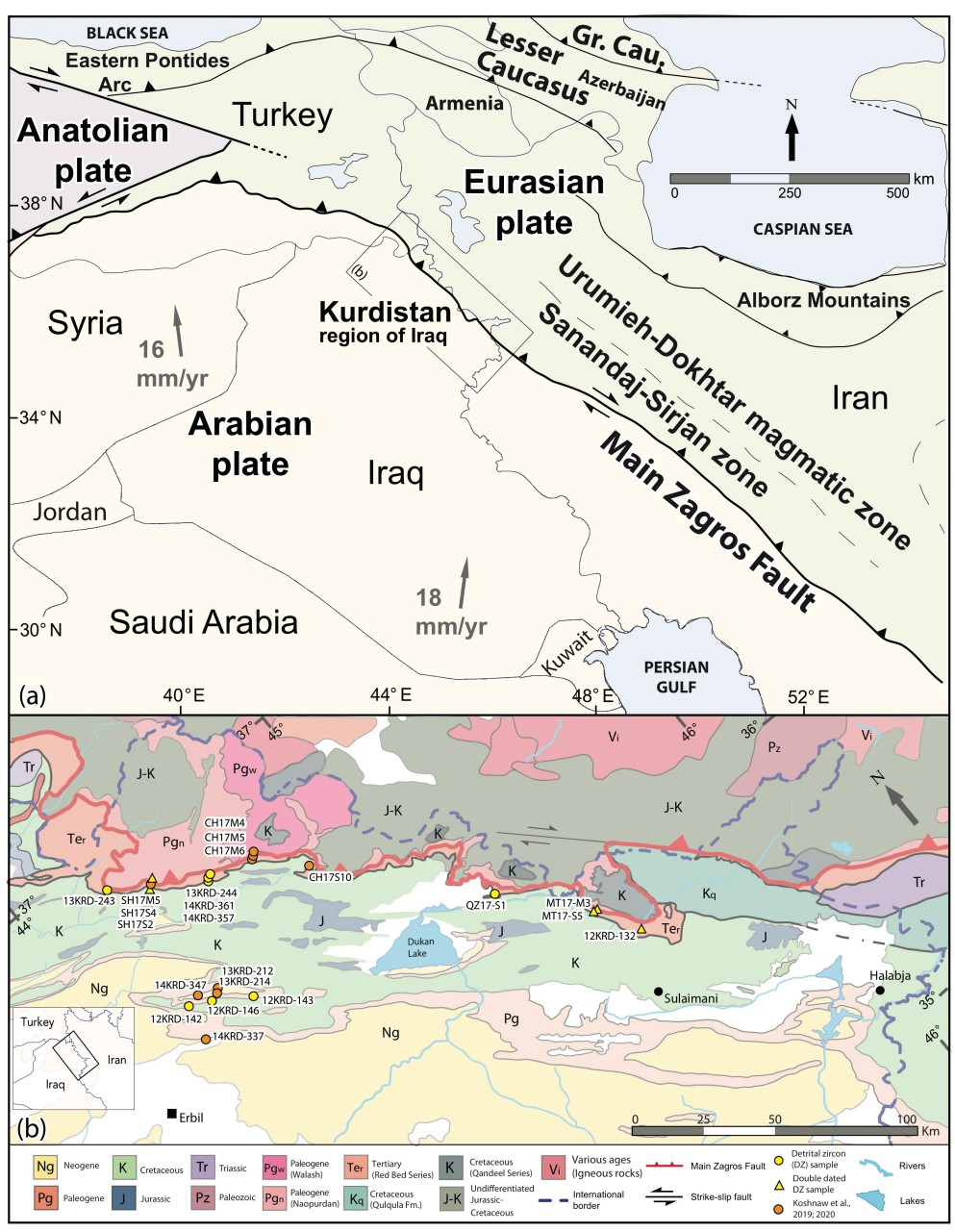


Figure 1. (a) Regional tectonic map of the Middle East displaying the study area, Kurdistan region of Iraq, and the suture of the Arabian and Eurasian plates along the Main Zagros fault (Koshnaw et al., 2017, and references therein). (b) Geologic map of the NW Zagros in the Kurdistan region of Iraq showing sample locations, new (yellow) and published (orange) (Koshnaw et al., 2019, 2020a, and references therein).

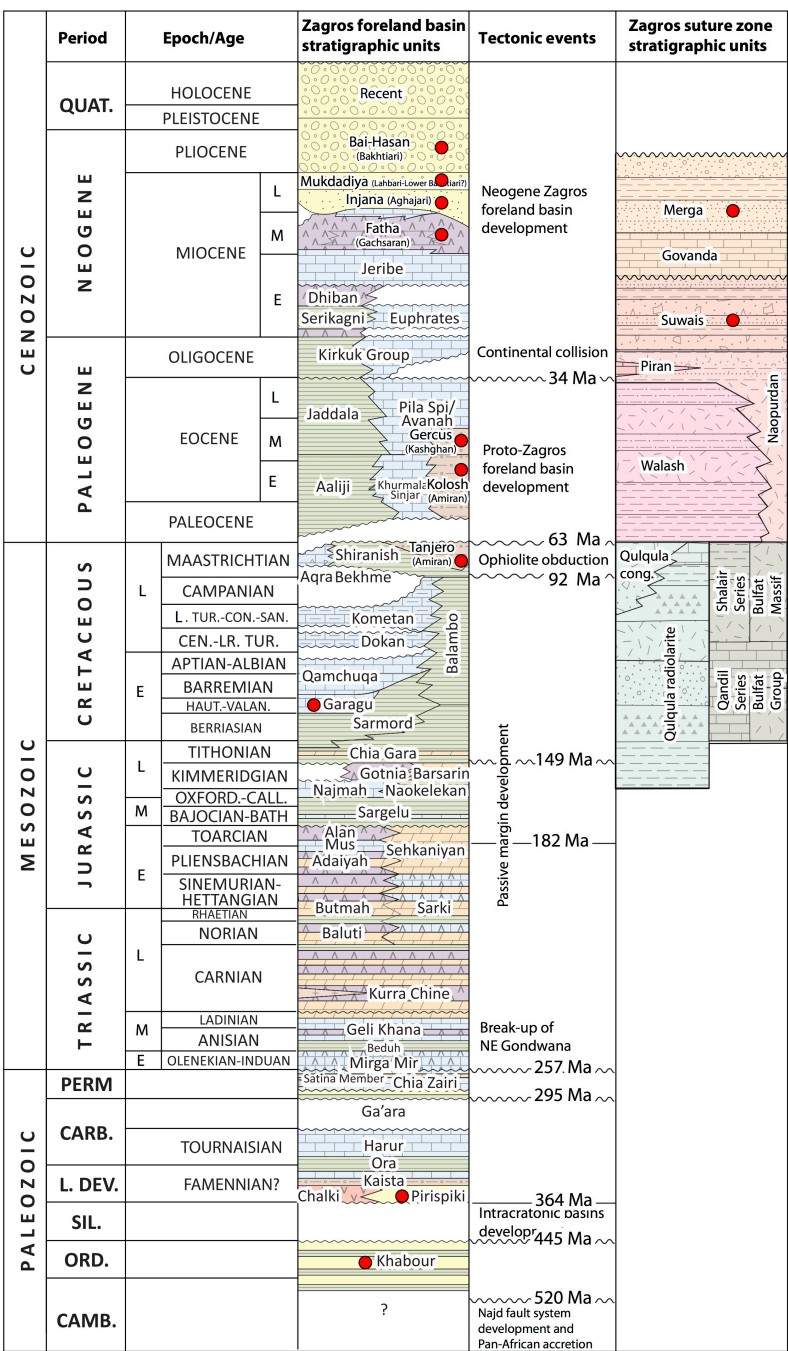

Figure 2. Generalized Phanerozoic stratigraphic column and key tectonic events for the NW Zagros fold-thrust belt and foreland basin, including the suture zone units in the Kurdistan region of Iraq (Sissakian et al., 1997; Sharland et al., 2001; English et al., 2015). The name in brackets represents the potential equivalent formation name in Iran. Red circles represent stratigraphic units that were addressed in this research.

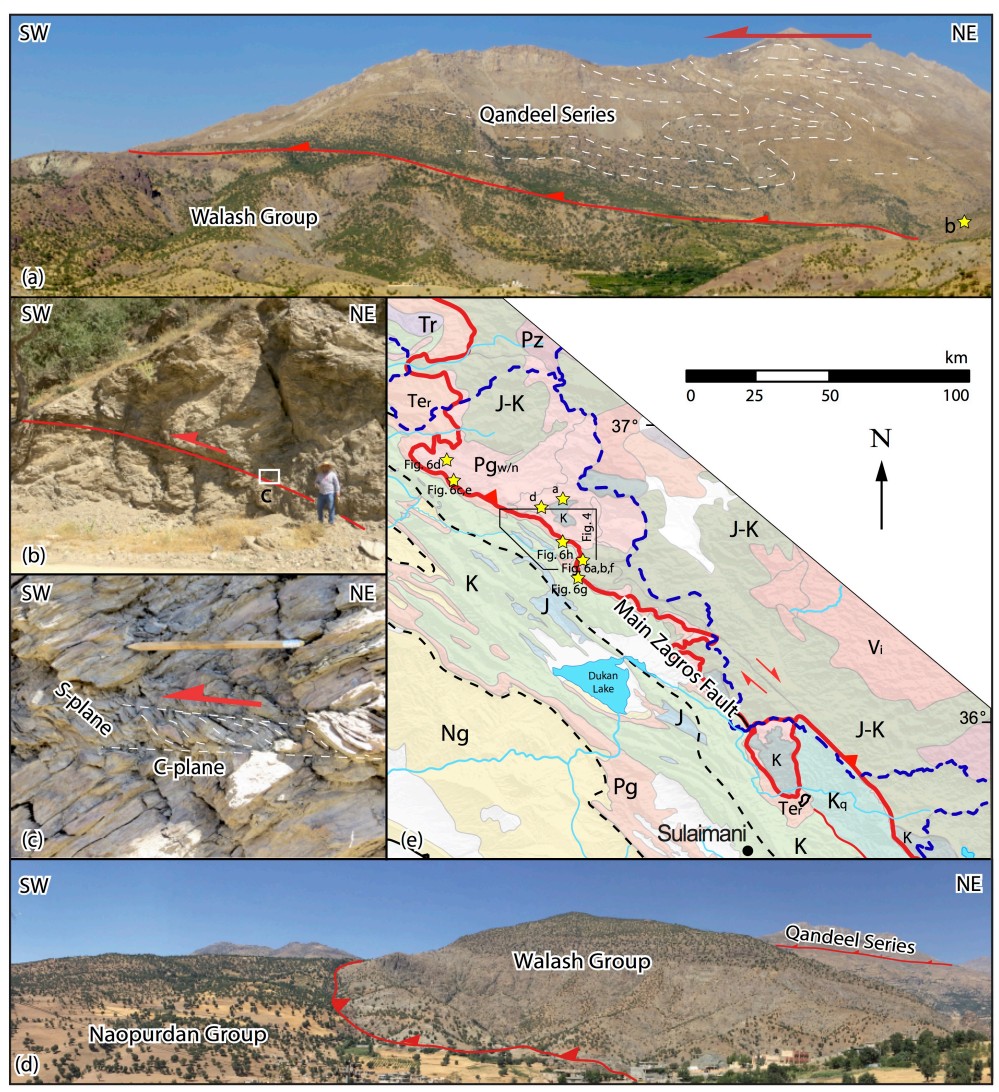

Figure 3. Landscape and outcrop photos exhibiting (a-d) main transported thrust sheets and their interpreted direction of motion (c, top-SW sense of shear) in the NW Zagros suture zone. (e) Simplified geologic map showing field photograph locations (yellow star).


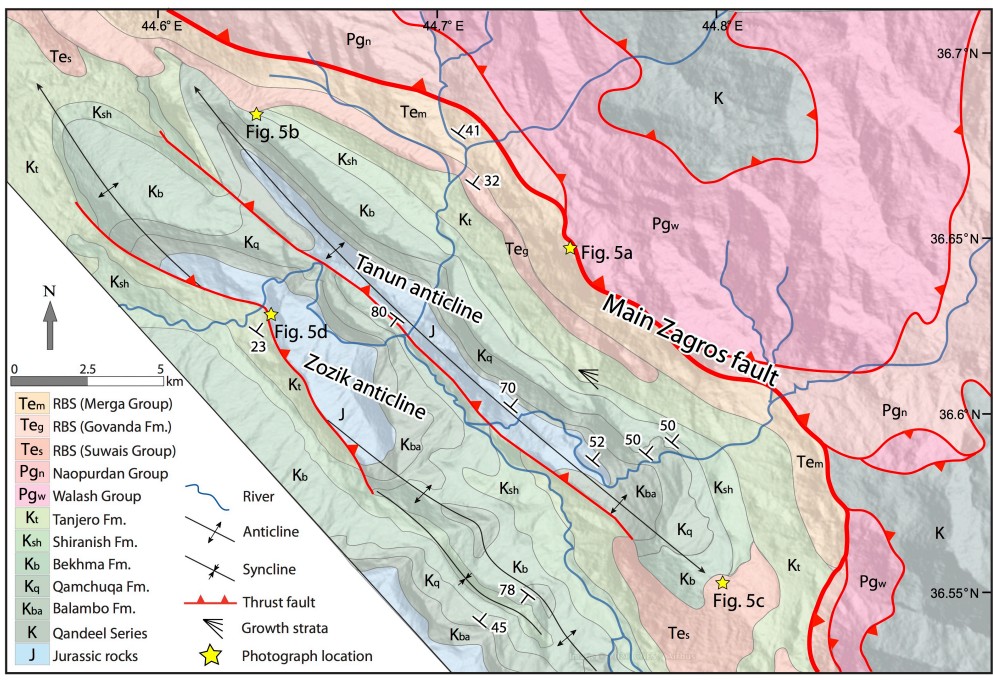

Figure 4. Geologic map illustrating key structural features, growth strata location of the Tanjero Formation (Le Garzic et al., 2019), and angular unconformity locations in the vicinity of the northwestern and southeastern plunges of the Tanun anticline. Base maps: Sissakian et al. (1997), MNR (2016); and Le Garzic et al. (2019).

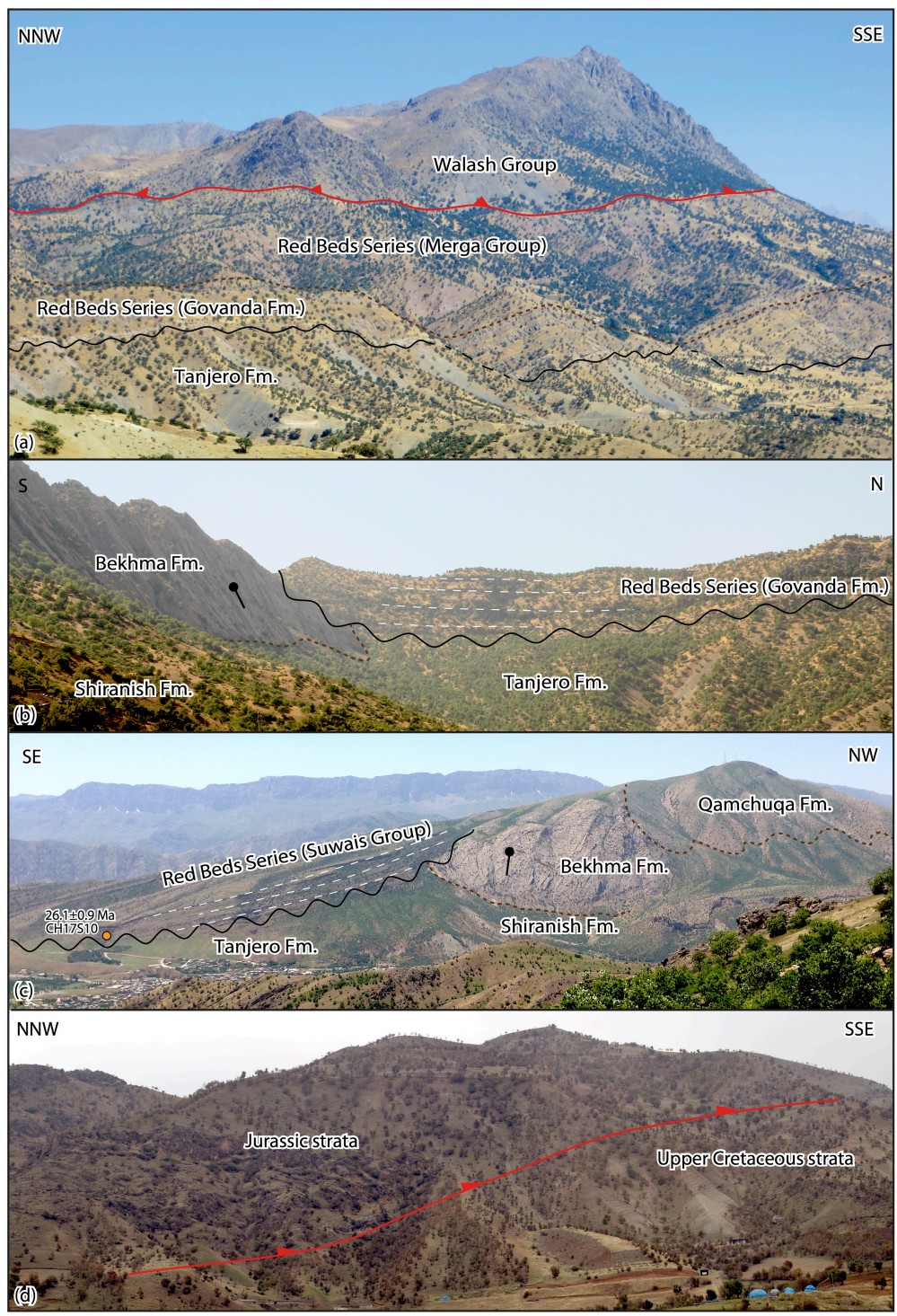


Figure 5. Landscape photographs showing field relationships. (a) The transported Paleogene Walash Group thrusted onto the Miocene units of the Red Beds Series that deposited on the Maastrichtian Tanjero Formation. (b) Angular unconformity between the middle Miocene Govanda Formation of the Red Beds Series and the folded Cretaceous units. (c) Angular unconformity between the upper Oligocene-lower Miocene (?) Suwais Group of the Red Beds Series and the folded Cretaceous units. (d) Thrust fault between the Lower
Jurassic strata (Sehkaniyan Formation?) and the Upper Cretaceous strata (Tanjero Formation).

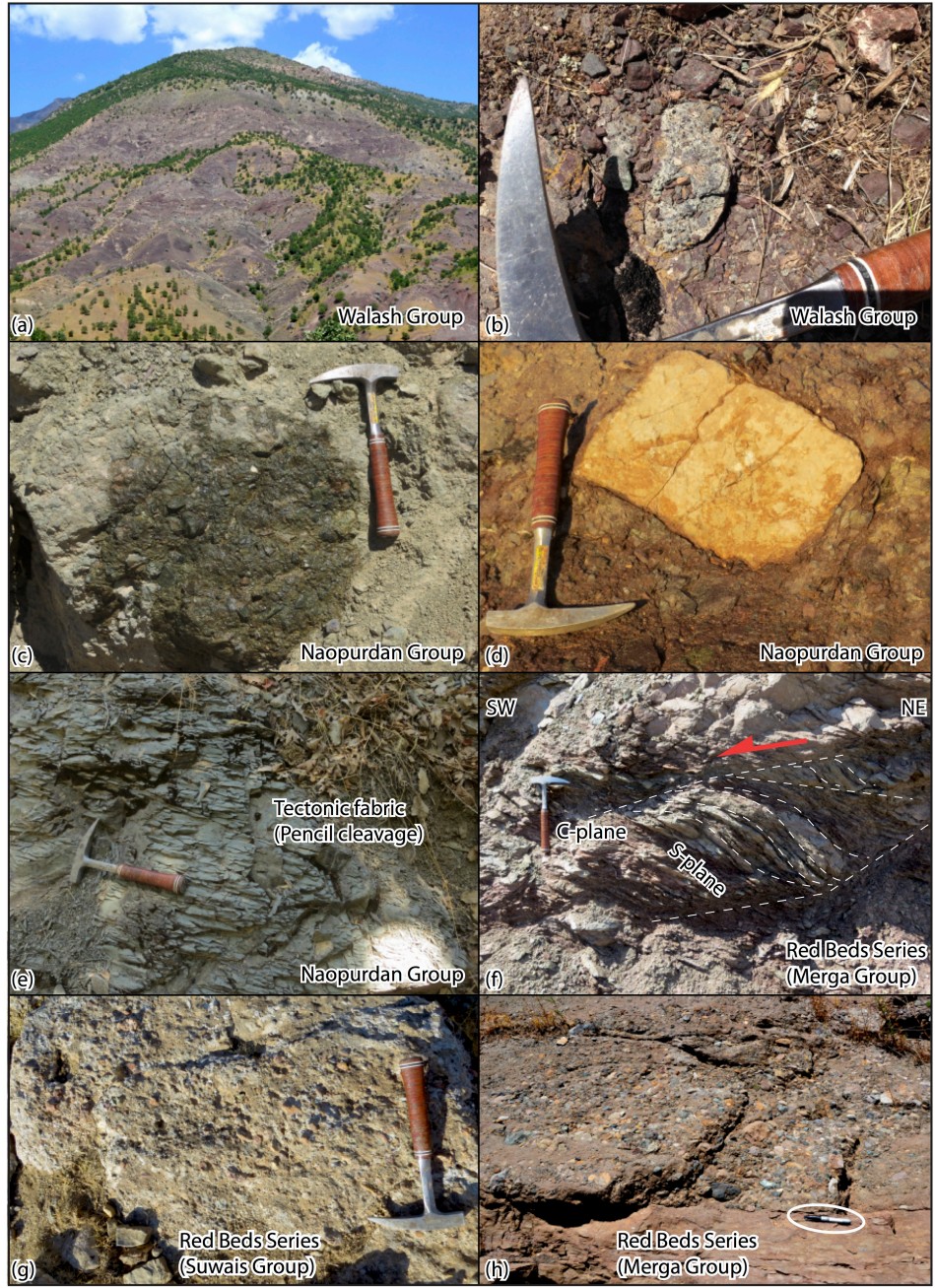

Figure 6. Landscape and outcrop photos displaying field examples of (a, b) Walash Group, (c, d, e) Naopurdan Group, Red Beds Series units of (f, h) Merga Group and (g) Suwais Group. The pencil cleavage tectonic fabric (e) developed in the lower part of the Naopurdan Group that thrusted onto the Merga Group. The C-S fabric developed in the upper part of the Merga Group that overlain by the thrusted Walash Group.


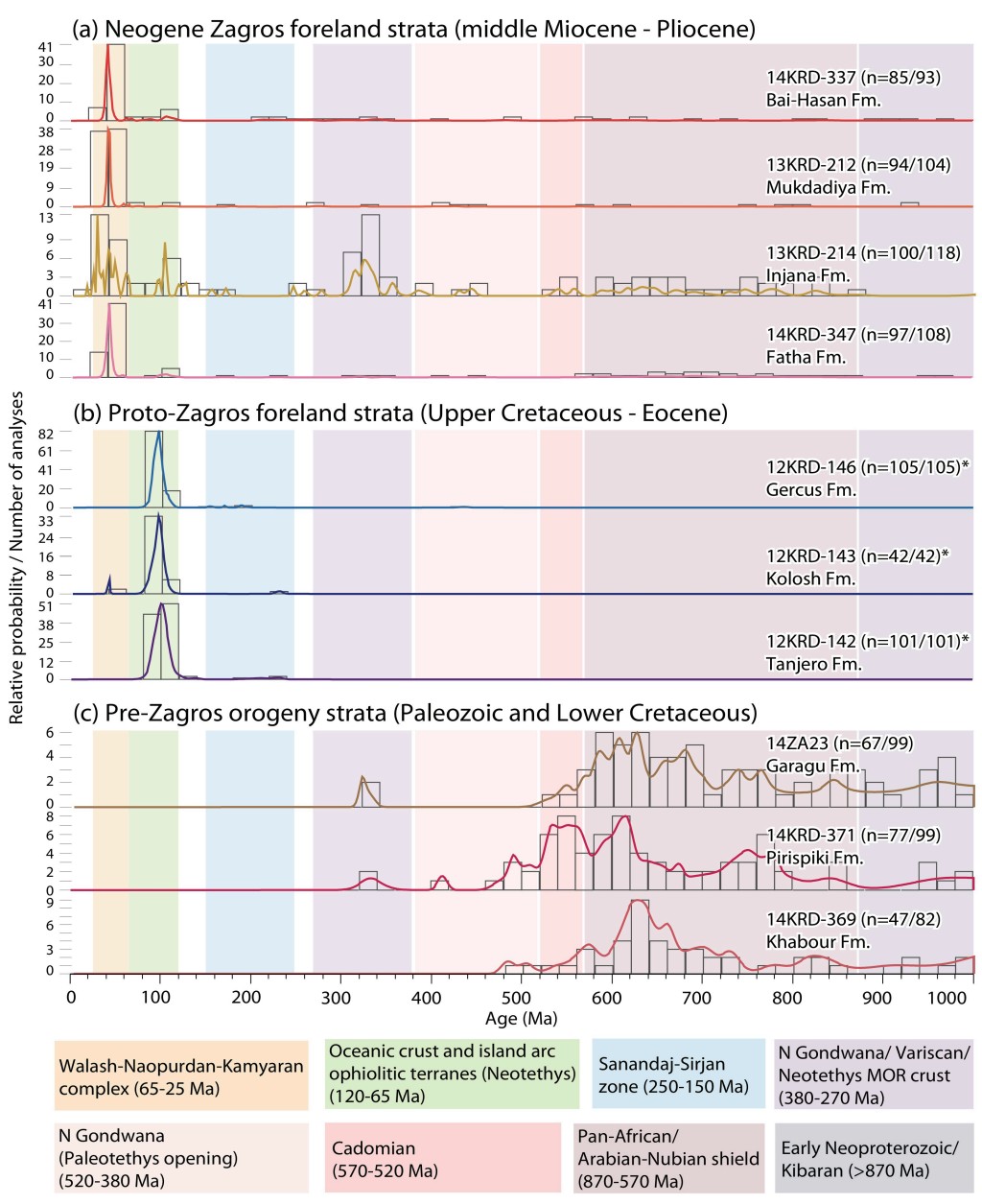

Figure 7. Detrital zircon U-Pb age distribution plots for the (a) Neogene Zagros foreland basin, (b) proto-Zagros foreland basin, and (c) pre-Zagros orogeny deposits depicted as probability density plot (PDP; bandwidth = 20, bin width = 20) and age histograms (Vermeesch, 2012). Color shading represents key source terrane age components. Samples with asterisk are new samples. Published samples are from Koshnaw et al. (2017 and 2020a). Sample locations shown in Fig. 1b.



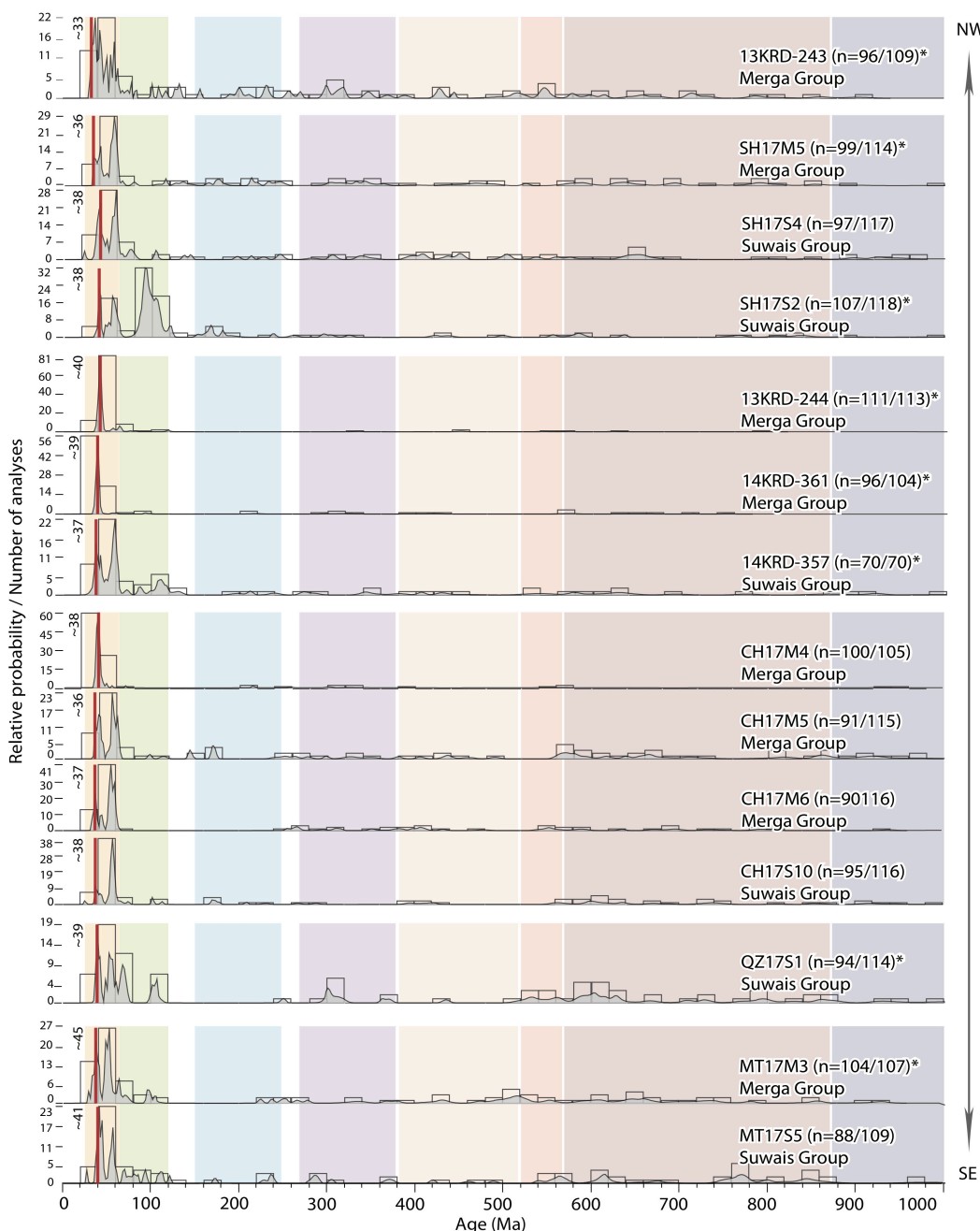

Figure 8. Detrital zircon U-Pb age distribution plots for the Red Beds Series units of the Suwais and Merga Groups depicted as probability density plot (PDP; bandwidth = 20, bin width = 20) and age histograms (Vermeesch, 2012). Dark red vertical bars delineate youngest age peak (≥3 grains). Color shading represents key source terrane age components, see Fig. 7 for legend. Samples with asterisk are new samples. Published samples are from Koshnaw et al. (2019). Sample sets are separated by a white space according to their localities. Sample locations shown in Fig. 1b.

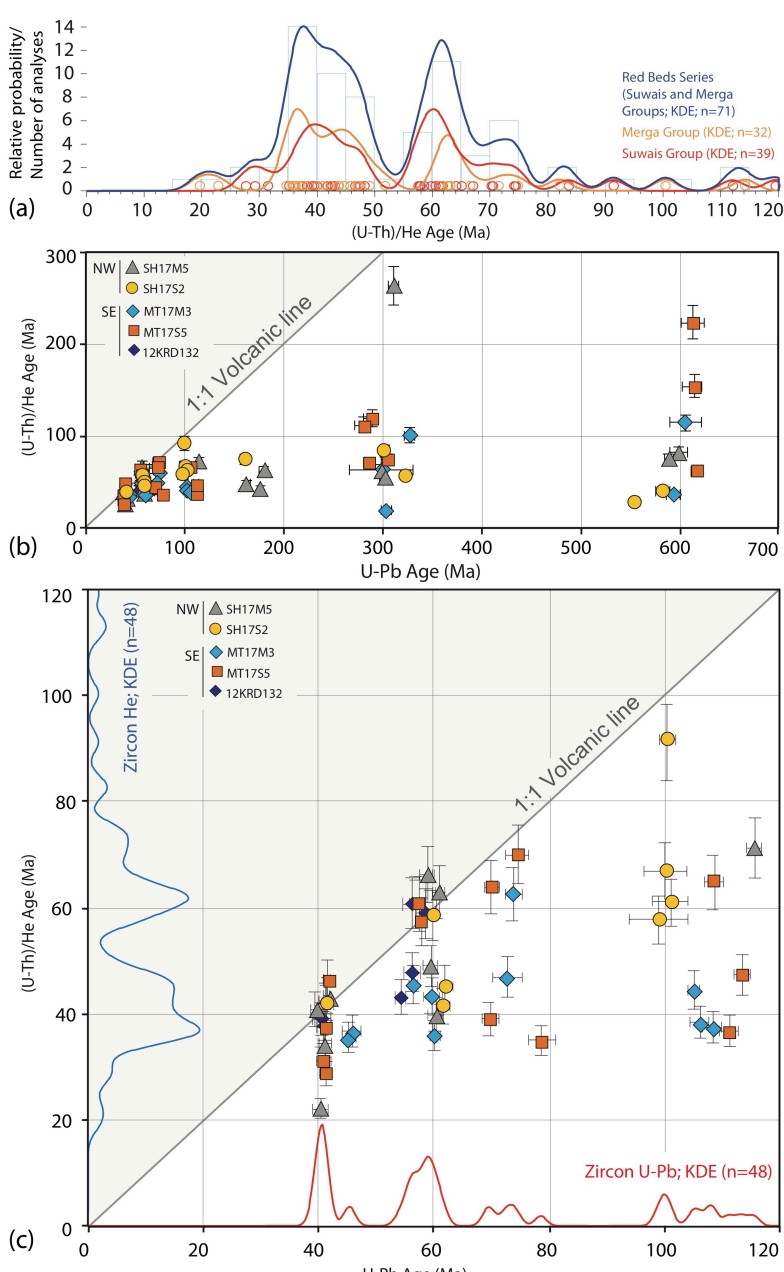

Figure 9. (a) Detrital zircon (U-Th)/He age distribution plots for the Red Beds Series units of the Suwais and Merga Groups depicted as kernel density estimation (KDE; bandwidth = 2, bin width = 5) and age histograms (Vermeesch, 2012). (b) Detrital zircon (U-Th)/(He-Pb) double dating plots of individual zircon analysis. (c) Expanded view of 0-120 Ma period to show detrital zircon (U-Th)/(He-Pb) ages for the Late Mesozoic and Cenozoic times.

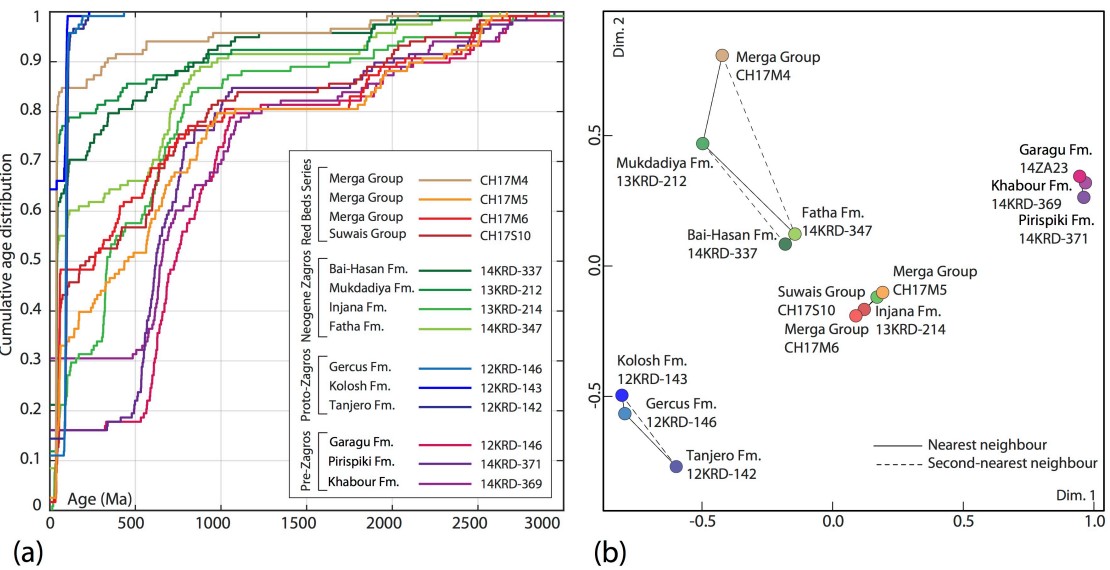


Figure 10. Comparison plots of the pre-Zagros orogeny, proto-Zagros foreland basin, Neogene Zagros foreland basin, and Red Beds Series deposits in the present-day Zagros hinterland. (a) Detrital zircon U-Pb cumulative age distribution (Saylor and Sundell, 2016). (b) Two-dimensional multidimensional scaling plot depicting relative statistical similarity/dissimilarity based on Komolgorov-Smirnov test D value (Vermeesch, 2013) of the detrital zircon U-Pb ages for the same sample sets in the cumulative age distribution plot. Dimension 1 and 2 are
unitless.

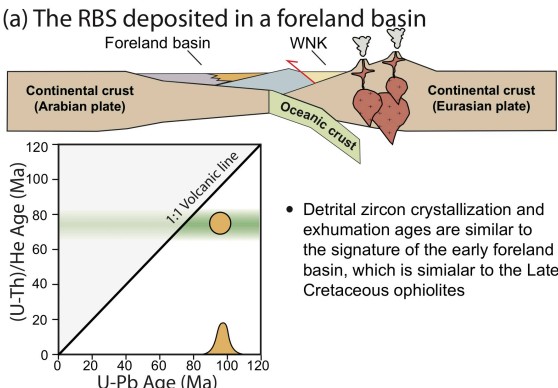

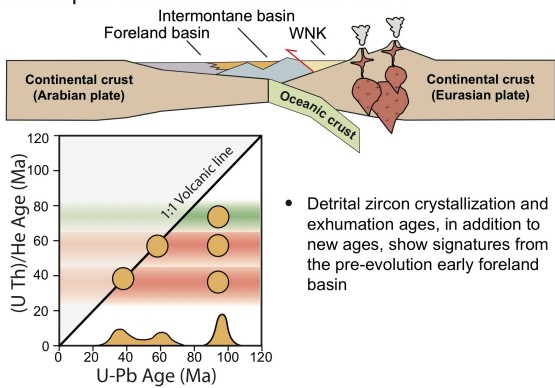

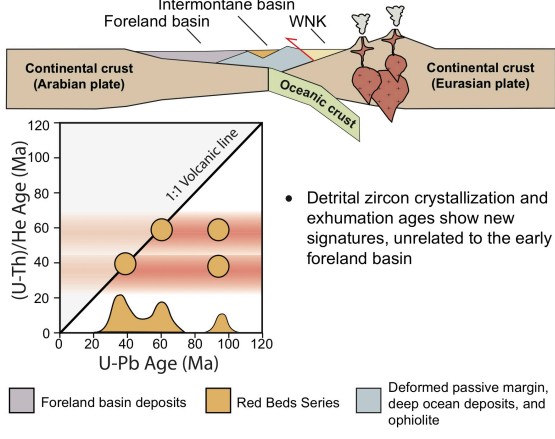

Figure 11. Predicted detrital zircon (U-Th)/(He-Pb) double dating ages for competitive hypothesis of the Red Beds Series depositional setting (a) a foreland basin, (b) a foreland basin, but later partitioned and evolved into an intermontane basin, (c) an intermontane basin.


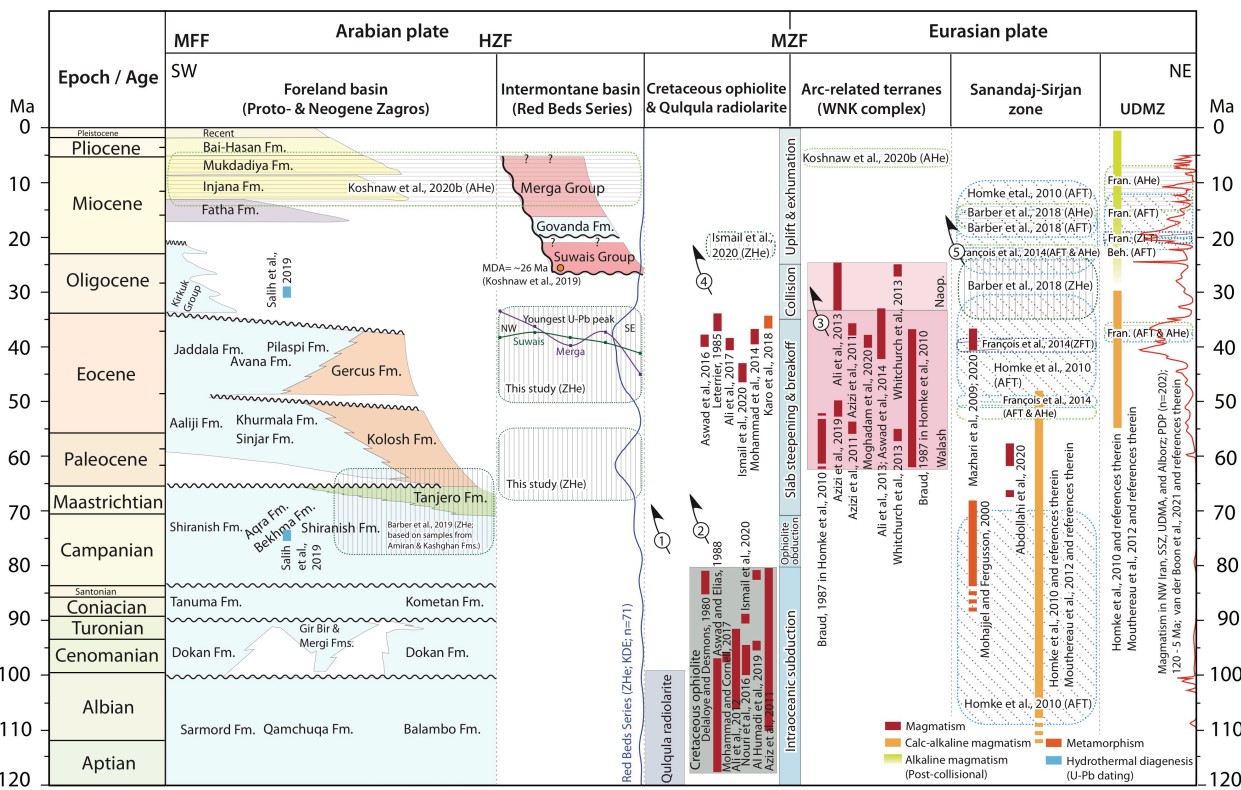

Figure 12. Time-space chart synthesizing hinterland basins stratigraphy and structural zones thermochronological and geochronological constraints on key tectonic and geodynamic, metamorphic, magmatic, and exhumational events in the broader NW Zagros sector. Numbered arrows represent estimated relative timing of thrusting of their respective terranes. Foreland basin stratigraphy based largely on van Bellen et al. (1959), Aqrawi et al. (2010), and Al-Husseini (2000). MDA, maximum depositional age; AHe, apatite (U-Th)/He; ZHe, zircon (U-Th)/He; AFT, apatite fission track; ZFT, zircon fission track; MFF, mountain front flexure; HZF, high Zagros fault; MZF, main Zagros fault; WNK, Walash-Naopurdan-Kamyaran; UDMZ, Urumieh-Dokhtar magmatic zone.



**(d) Miocene**

- Deposition of the RBS
- Renewed thrusting of the WNK complex onto the RBS
- Out-of-sequence thrusting, carrying segments of
  the Upper Cretaceous ophiolite, including the Eucene intrusions,
  onto the WNK complex
- Development of the Neogene Zagros fold-thrust belt and foreland basin
- Crustal thickening, slab breakoff, and uplift
- Increase of post-collisional magmatism

**(c) Oligocene**

- Initial collision potentially after ~36 Ma
- Terminal collision potentially by ~26 Ma
- Deposition of the RBS in an intermontane basin
  in the hinterland of the proto-Zagros fold-thrust belt
- Thrusting of the WNK complex onto the RBS
- Reduction of magmatism

**(b) Eocene**

- Potential slab breakoff near
  Arabia, and slab rollback and
  flattening beneath Eurasia
- Magmatism in the Upper Cretaceous
  ophiolite, WNK complex, SSZ, and UDMZ

**(a) Latest Cretaceous-Paleocene**

- Emplacement of the Neotethys ophiolitic terranes onto Arabia
- Development of the proto-Zagros fold-thrust belt and foreland basin
- Development of the Walash-Naopurdan-Kamyaran complex

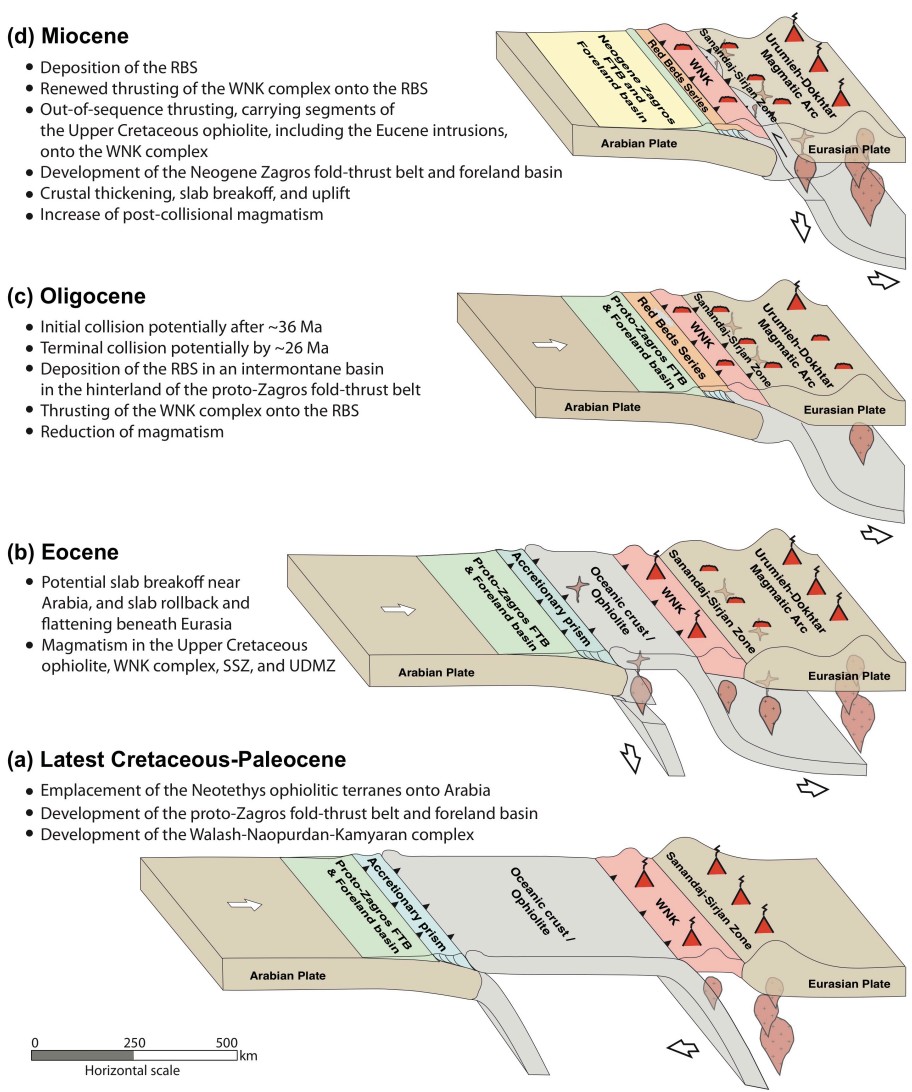

Figure 13. Schematic illustrations portraying the Late Mesozoic-Cenozoic history of the Arabia-Eurasia convergence and the NW Zagros suture zone evolution. The possible configuration of the subducted Neotethys oceanic slabs, and the Arabian and Eurasian plate margins are constrained by results of this study and previous data and interpretations (Agard et al., 2005; Barrier and Vrielynck, 2008; Omrani et al., 2008; Agard et al., 2011; Moghadam and Stern, 2011; Mouthereau et al., 2012; Whitechurch et al, 2013; Ali et al., 2014; Jagoutz et al., 2015; Hassanzadeh and Vernicke, 2016; and references in Fig. 12).

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

**Competing interests**. The authors declare that no competing interest is present.

**Code/Data availability.** The new data used in this study will be available on Mendeley Data.

**Author contribution.** Renas Koshnaw: Conceptualization, investigation, data collection, writing original draft, reviewing and editing, visualization, funding acquisition. Fritz Schlunegger: Investigation, resources, funding acquisition, reviewing and editing. Daniel Stockli: Investigation, data collection, resources.
