# Peer review of "Detrital zircon provenance record of the Zagros mountain building from the Neotethys obduction to the Arabia-Eurasia collision, NW Zagros fold-thrust belt, Kurdistan region of Iraq"

_Solid Earth, 2021_

## Author Comment (AC1)

**General comments**

This paper presents 1097 new U-Pb ages and 74 new AHe on zircon providing 74 double dates on key deposits from the Zagros foreland and proto-foreland and sediments from the Zagros (Kurdistan) suture zone. These constrains are used to position the Red Beds Series and decipher between different scenarios of the evolution of the transitional domain stretching between the Arabian margin and the active margin of Eurasia. The paper is generally well written and organised expect the introduction that requires shortening and rewriting. This part also contain several unclear statements (see below). My main concern is about the discussion. The discussion indeed implements models of large scale geodynamic reconstructions but unfortunately do not enough present similar constraints (stratigraphy, basin evolution and sparse thermochronological constraints or description of sediments) obtained in other regions of the Zagros including Kermanshah region but also in the Fars. As presented below there are several lines of evidence suggesting the Red Beds Series and the WNK volcanic complex could correlate with deposits identified along the strike of Zagros suture zone. This has obvious implications on the architecture of the suture zone that could argue for a cylindrical margin over more than 2000 km. Also I suggest the Red Beds Series may be part of the Zagros foreland above the obducted units and above the the proto-foreland basin.

Thank you for your comments. The introduction section has been edited and shortened. Potential equivalent formation names in Iran have added. The Kermanshah region was considered in the context of the Lurestan segment, but for this revised version the Dezful and Fars regions were also included, especially for the presence of the Razak Formation. The concern about the RBS being part of the collisional foreland basin has also been addressed (see details in the next section). We agree that the RBS might have been deposited on top of the obducted ophiolite and the proto-foreland. The geodynamic model figure (13c, d) has been updated to show this aspect more explicitly.

**Main comments**

The U-Pb/Zhe age signature obtained for RBS may be compared to the few AFT data from the base of the Neogene strata (Razak Fm) of the Fars region of the iranian Zagros (Khadivi et al., 2012) revealing source from a mixture of ophiolitic series (100 Ma) and arc derived rocks (magmatic or erosion events in the range 66-39 Ma) that was emplaced in the inner part of the Zagros and likely covered part of the High Zagros. In this work the base of the foreland deposits (Razak FM), which is found only to the north close to the MZT, share some FT ages (but not U-Pb ages signature) with the early (proto-)foreland deposits of Paleogene age (e.g. Amiran Fm).

In the scenario presented the WNK complex would be a possible source of Razak Fm that we never found preserved in the Fars. It terms of terminology and spatial correlation would it be possible that the RBS=Razak Fm ? In this case the RBS could be part of an inner foreland basin as commonly described in the Iranian Zagros rather than intermontane basin. In other word, the zagros suture zone was the inner foreland during the Neogene.

In addition, an erosional event is described in the Eocene in the high Zagros (Mouthereau et al., 2012; Khadivi et al. 2012) in the Lorestan and the Fars. This could correspond to the

unconformity inferred at the base of the Suwais Fm. Although the margin of Arabia was obviously variable along strike such equivalence would make the foreland more cyclindrical.

There is a possibility that the RBS and Razak Formations are equivalent, but may vary in facies depending on where they had been deposited. See below new texts from the revised version of the manuscript:

The targeted NW Zagros hinterland deposits are the Red Beds Series (potential equivalent of the Razak Fomration in Iran; Etemad-Saeed et al., 2020), below the Main Zagros fault, and the proto-Zagros Tanjero (Amiran), Kolosh (Amiran), and Gercus (Kashghan) Formations in the Kurdistan region of Iraq (Fig. 1b).

Along-strike of the Zagros suture zone toward Iran, the potential equivalent of the RBS, the Razak Formation, appears not to be preserved in the Lurestan segment, but in the Dezful and Fars segments it has been documented (James and Wynd, 1965; Alavi, 2004; Khadivi et al., 2010, 2012; Vergés et al., 2018; Etemad-Saeed et al., 2020). The Razak Formation shows a lithostratigraphy and a possibly timing of deposition that is comparable to the RBS particularly in the hinterland. The stratigraphy includes carbonates and calcareous argillites toward the foreland and shales, siltstones, sandstones and conglomerates toward the suture zone (Alavi, 2004). Available magnetostratigraphic dating, yet between the High Zagros fault and the Mountain Front Flexure and not adjacent to the Main Zagros fault, suggest 19.7 – 16.6 Ma as the time of deposition of the Razak Formation (Khadivi et al., 2010). The older deposits of the Razak Formation closer to the Main Zagros fault might have been eroded due to deformation and exhumation (Alavi, 2004; Khadivi et al., 2012). These characteristics of the lithology and the timing of deposition appear to resemble those of the Govanda Formation and the Merga Group of the RBS. However, unlike the RBS in the Kurdistan region of Iraq, the Razak Formation seems to be more wide spread geographically from the hinterland toward the foreland and in direct contact with the Asamri (Jeribe), Gachsaran (Fatha), and Aghajari (Injana) Formations (Alavi, 1994; Khadivi, 2010; Etemad-Saeed et al., 2020).

This conclusion about the RBS in the NW Zagros in the Kurdistan region of Iraq can also be considered for the Razak Formation in the Dezful and Fars areas of Iran, but with a possible variation regarding the structural architecture of the basin. In the study area, the RBS is structurally bounded by the allochthonous WNK complex toward the NE is and by the anticlines of the proto-Zagros fold-thrust belt toward the SW (e.g. Figs. 4 and 5), and it is in direct contact with the Paleogene and older rocks. However in the Dezful and Fars segments of Zagros, where the development of the Paleogene proto-Zagros fold-thrust belt is limited (Hessami et al., 2001), the basin setting of the Razak Formation appears to be less restricted geographically, and the sediments appear in direct contact with the Neogene Zagros foreland basin deposits with no documented unconformities (James and Wynd, 1965; Khadivi et al., 2010; Vergés et al., 2018; Etemad-Saeed et al., 2020). Such a basin setting for the Razak Formation could facilitate a more straightforward interpretation as early foreland basin deposits during the Neogene. In either case, both the RBS and the Razak Formation are representing the deposits of the early stage

collision between the Arabia-Eurasia plates (Khadivi et al., 2012; Koshnaw et al., 2019; Etemad-Saeed et al., 2020).

Moreover, in the southeastern Fars segment of the Zagros belt, the AFT detrital age form the potential equivalent of the RBS, the Razak Formation, yielded ~25 Ma, and it has been interpreted to record the exhumation of the Main Zagros hanging wall in the suture zone (Khadivi, et al., 2012). Such a similarity in the suture zone exhumation during the latest Oligocene and the earliest Miocene across the Zagros ~2000 km-long orogen may suggest a uniform suturing between Arabia and Eurasia, and possibly reduction of the oblique collision effect and onset of the northward motion of Arabia (McQuarrie et al., 2003; Navabpour et al., 2013).

**Specific comments**
This part is unnecessary long and wordy. It contains disconnected sentences. I suggest to shorten in 2 short sentences.
The text has been shortened as follow: Hinterland basins, such as wedge-top and intermontane basins are valuable archives for the assessment of the exhumation and unroofing history of the adjacent uplifted terranes because of their proximity to the source areas. Nevertheless, well-preserved ancient stratigraphic successions are scarce due to the deformation of sedimentary strata as orogenesis proceeds (Horton et al., 2012; Orme et al., 2015). A possible approach to overcome this drawback is the utilization of geochronologic and thermochronologic records, which are preserved by detrital zircons (e.g. Cawood et al., 2012; Webb et al., 2013; Gehrels, 2014; Colleps et al., 2020).

L55 : No. The Zagros orogen did not form as a result of obduction but due to convergence and most likely collision between Eurasia and Arabia plates.
The text has been edited as follow: This orogenic belt formed during the Late Cretaceous and Cenozoic as a consequence of the Arabia-Eurasia convergence and their collision

L58: Different terranes ? which ones ?
L59: Why uncertainties arise from the almagation of different tectonic terranes ? If their geometry and kinematics are simple then the reconstructions can be straightforward.
The text has been edited as follow: This prolonged history of deformation resulted in an amalgamation of deferent tectonic terranes between the Arabian and Eurasian plates, such as the Bisotoun block, the middle Cretaceous intraoceanic oceanic subduction and back-arc spreading zone, the early Tertiary magmatic domain, overprinting the preceding tectonic configurations (Wrobel-Daveau et al., 2010; Agard et al., 2011; Vergés et al., 2011; Barber et al., 2018, 2019).

An example for uncertainty, defining the length and width of the stretched Arabian margin prior to collision, which might influence the timing and the style of the hard collision.

L61-62: … Walash Fm.. Red Beds Series. Not yet introduced.
They have been removed.

L108-109: Yes but not only. Acknowledge also older works.

New references have been added (Stoneley, 1975; Koop and Stoneley, 1982; Alavi, 1994)

By the Maastrichtian time the proto-Zagros flexural foreland basin started to form in response to the arrival of the Neotethys intraoceanic subduction zone at the Arabia plate margin, leading to the ophiolite obduction (Stoneley, 1975; Koop and Stoneley, 1982; Alavi, 1994; Homke et al., 2010; Saura et al., 2011; Barber et al., 2019).

L118-121: how do these successions relate to the deep marine to shallow marine transition you mentioned above ?

The text has been edited and the text about the Fatha (Gachsaran) Formation was moved to the next section in the text.

On top of the Pila-Spi Formation, an unconformity has been recorded based on absence of the Oligocene-early Miocene rocks that are linked to the Arabia-Eurasia collision (Fig. 2) (Dunnington, 1958; Ameen, 2009; Lawa et al., 2013).

L127: Would be useful to know how these formations correlate with more familiar stratigraphy of the Iranian Zagros. This could be done in Figure 2.

The potential equivalent formations have been added in the text and Fig. 2.

L235: This is not true. The neogene period is obviously syn-collision.

Edited to be "collision-related"

L236: Tanjero Fm belongs to the obduction phase according to your stratigraphic chart not to the proto-Zagros. This is a lot of names. Wherever possible add the stratigraphic ages.

The stratigraphic ages and potential equivalent formations have been added.

In particular, the provenance data from the Maastrichtian Tanjero (Amiran), the Paleocene-Eocene Kolosh (Amiran), and the Eocene Gercus (Kashghan) Formations

L239: The WNK complex should be defined earlier when you introduce the Walash-Naopurdan series for the first time.

The WNK has been defined immediately after description of the Naopurdan Group.

The Walash and Naopurdan Groups were correlated with similar rock units in the adjacent part of the Zagros belt in Iran and named as Walsh-Naopurdan-Kamyaran (WNK) (Ali et al., 2014; Moghadam et al., 2020).

L257: It is not trivial to associate zircon with the expected juvenile composition of magmas on mid-oceanic ridges. the same comment holds when you relate zircon U-Pb age of 100 Ma with the Tethys. Although common in cumulate this is not expected in basalt. Just add a few words perhaps in Chapter 2, to make this point clearer.

L258: Somewhat related to my comment above but this point should address earlier when you present the geology of potential sources.

We agree. The comparison and the age similarities are based U-Pb ages for zircons. The text has been edited as follows: Even though zircon is not a common mineral in the mafic rocks, yet it has been recognized (Grimes et al., 2007 and references therein). The ~240 Ma Triassic age signature in the Amiran and Kashghan Formations has been attributed to the mid-oceanic ridge based on the zircon trace element data (Barber et al., 2019).

L262: Unclear why you need to recycle sediments of this basin which is rich in carbonates.

The sentence has been edited as follow: Such variation in the sediment source for the proto-Zagros foreland basin is in line with the destruction of the Gotnia basin architecture and the input of carbonate materials into the newly formed flexural basin, and the occurrence of some recycled Paleozoic and older zircon grains in the Tanjero Formation

L326-327: This is not what is shown in Fig. 5c.

Fig. 5c. Shows that the RBS (Suwais Group) deposited on the older Upper Cretaceous rocks. The detrital zircon data show similarity of provenance among different samples of the RBS, including the Suwais Group. So, we think and are convinced that it is indeed the case.

L339: Ok there is one finally. So why this is different from Le Garzic et al. 2019 ?

It seems that the Le Garzic et al. considered the RBS to be equivalent to the Kolosh (Amiran) Formation and possibly to the Gercus (Kashghan Fromation) that were deposited during the Paleocene and the Eocene. Such view is common in most of the papers that are dealing with the Zagros belt in Iraq. However, this manuscript considers the onset of the RBS deposition to be of a late Oligocene age. This age assignment is based on a maximum depositional age recorded by detrital zircon minerals (Koshnaw et al., 2019). Additionally, as discussed in the discussion section *Basin dynamic of the NW Zagros Red Beds Series deposits*, the U-Pb and the ZHe data do not support the occurrence of a single basin especially during the Paleocene-Eocene.

L353: But what if RBS have been originally deposited above WNK which series were later emplaced during the Neogene ?

This is unlikely because there is field observation that the RBS was deposited on strata of the Upper Cretaceous Arabian plate (e.g. Figs. 4 and 5). Additionally, so far no thrust fault has been identified between the RBS and the other Arabian plate strata.

---

## Author Comment (AC2)

**General comment:** The manuscript presents new detrital zircon age data and intends to combine the data with field observations to conclude that unlike the existing plate tectonic model for Iraq the Paleogene Walash-Naopurdan-Kamyaran arc-related complex formed on the Eurasian side of the Neotethys. But the presented new model is poorly substantiated and the targeted rock units for detrital zircon study is not satisfactory. Based on the comments summarized below, my conclusion is that the current manuscript has been prepared in a rush and needs a major repair before being considered for publication.

* Thank you for your comments. The geodynamic scenario presented in this manuscript accounts for (i) the new data presented in this work, (ii) previously published data, (iii) the existing magmatic and tectonic constraints, and for (iv) the available plate kinematic models for the NW Zagros, including Turkey, Iraq, and Iran. Whenever there is new information, any model can be revisited. A tectonic model for the NW Zagros should be valid for the entire area and beyond the political boundaries of countries.

**Specific comments:** Data: Introducing of the situation of the "new data" is not clear and data tables are missing. Lines 70-74 say that the new provenance information will be combined with published data. Then there is a mention of "five" samples which have been selected for detrital zircon double dating. Till here, the reader learns that new data for five samples are being used in this paper. Later, Lines 185-189 state that 1097 new detrital zircon U-Pb ages and 74 new detrital zircon helium ages are presented from "eigh"t Red Bed Series samples and "three" samples from the proto-Zagros formations. But no data table is attached to check all that. In the beginning, I thought perhaps I have missed downloading the Tables but then I realized that no reference has been made to a data table in the text. By piecemeal search throughout the MS, I found a general picture of the data source in Figure 12, oddly referred to only in Figure 13. This Figure 12 nicely presents a summary of the previous work and the current study linked with an interpreted stratigraphy but strangely this figure is not cited properly throughout the text.

* FYI, just before the "five" there is "Furthermore" and immediately after the "double dating" there is "to tune-up the link…". These are key words that indicate the detrital zircon double dating analysis is extra work on top of the detrital zircon U-Pb work that is mentioned in the beginning of the paragraph. For the purpose of a better clarification the text has been edited as indicated below, and the number of samples and analysis were moved to the *Sampling and methods* section.

* Introduction: This research aims at constraining (i) the basin dynamic recorded by the suture zone deposits and the wedge-top units, and (ii) the Arabia-Eurasia convergence history based on the detrital zircon U-Pb and (U-Th)/(He-Pb) double dating methods.

* Sampling and methods: In this paper 1097 new detrital zircon U-Pb ages are presented from 11 samples (Supplemental tables 1 and 2), eight samples from the Red Beds Series and three samples from the proto-Zagros formations. These new data are integrated with previously published U-Pb ages in the study area (Koshnaw et al., 2019; Koshnaw et al., 2020a). Additionally, for the purpose of the zircon (U-Th)/(He-Pb) double dating, 74 detrital zircons were selected for conducting new (U-Th)/He analyses from these geochronologically dated

grains. These minerals were extracted from five Red Beds Series samples (Supplemental tables 1 and 3).

* The supplementary data tables were actually provided, please see the screenshot below, but yes the citation of the tables was missing in the text.

**Status: Discussion (SE Discussions)**

Assets for review 🔍
**Data sets**

| | |
|---|---|
| Title: | Supporting_information_Koshnaw_etal_SolidEarth |
| Authors: | Koshnaw et al. |
| Access restrictions: | Access limited to reviewers |

* Figures 1 and 2 show the samples location, type of analysis, new and published samples, as well as their stratigraphic location. Please look at pages 13 and 14 of the early version of the manuscript. Figure 12 is associated with the geodynamic reconstruction in the last section of the discussion. This is the reason why it is not cited earlier.

**Expression:** For someone not much into the stratigraphy of Iraq the text is hard to follow. Keeping names of such many rock units in the right age order in mind is no easy. It would help if age could be mentioned before the Formation name; for instance, "the Oligocene Swais Group" rather than just "Swais Group". Regarding the English of the text, I noticed frequent missing verbs, wrong verbs, and typos. That indicates the text has been submitted before comprehensive editing by the team of authors.
* The manuscript has been edited to address these issues, including the language, however the Red Beds Series and its individual units were described in detail in the section 2.4. Other formation names have been edited to be associated with their age and potential equivalent formation names in Iran as deemed necessary. Additionally, Fig. 2 shows the formation names and their respective ages.

**Tectonic model:** Koshnaw et al. propose a new model of tectonic accretion of the Tethyan Paleogene blocks onto the Eurasian side of the Neotethys. Their suggested model competes with an existing model which considers the pre-Miocene accretion happened on the Arabian side of the ocean. Contrasting with the conventional model (e.g., Aswad et al., 2014; Ali et al., 2019; Jones et al., 2020) formation of the WNK complex is now proposed to have taken place entirely on the Eurasian active margin. However, the supporting discussion for the new model is inadequate. For instance, development of the Paleogene WNK arc-related complex in juxtaposition with the Sanandaj-Sirjan zone requires that sediments within the former to be containing Triassic-Jurassic-Cretaceous age detrital zircons from the latter. Discussion of such aspects of the proposed reconstruction is missing.

**Methodology:** Since the main objective of the current manuscript is to show that the Paleogene arc activity along the WNK took place in the same tectonic setting as the Sanandaj-Sirjan arc, study of the detrital zircon content of the WNK complex rocks is required. Characteristic zircon U-Pb ages of the Sanandaj-Sirjan zone with conspicuous peaks for Ediacaran, Carboniferous-Permian, Triassic and Jurassic periods if seen in the WNK sediments would support the proposed tectonic model.

\* The goal of the manuscript is to utilize the detrital zircon provenance record of the Arabian plate to deduce the convergence history between Arabia and Eurasia as indicated by the title, introduction, and discussion sections.

\* The manuscript proposes a genesis of the WNK adjacent to the Eurasia. Not writing what part of the discussion, and for what reason, is inadequate, and it does not help making an argument. The detrital zircon geochronologic and thermochronologic data from the RBS suggest an origin from the WNK complex. The RBS detrital zircon record does contain the mentioned age components. Additionally, the majority of the double dated detrital zircons from the RBS that have the Ediacaran, Carboniferous-Permian, and Jurassic ages show a similar exhumation age of the Late Cretaceous to Eocene, consistent with the exhumation age of the Sanandaj-Sirjan zone (e.g. Homke et al., 2010; Khadivi et al., 2012; Mouthereau et al., 2012; Barber et al., 2018). The assumption of more than two subduction zones between Arabia and Eurasia during the Eocene faces a space problem, especially if we consider that collision initiated in the late Eocene. This aspect has been discussed in detail in the *Geodynamic evolution* section. Furthermore, the equivalent complexes in Turkey (Maden-Hakkari complex) and in Iran (Gaveh-Rud domain /Early Tertiary magmatic domain/Kamyaran Paleocene-Eocene complex) are all suggested to occur adjacent to Eurasia (Braud and Ricou, 1975; Yılmaz, 1993; Robertson et al., 2007; Oberhänsli et al., 2010; Homke et al., 2010; Saura et al., 2015; Agard et al., 2005, 2011; Whitechurch et al., 2013). Lastly, considering the Walash-Naopurdan complex close to the Arabia does not fit the broader plaeotectonic of the Middle East. No plate kinematic reconstruction support a setting for the WN that is different from the adjacent equivalent Paleogene blocks, as a different setting would cause unnecessary geometrical complication for the movement of the rigid blocks on a spherical surface within a limited space (~300 km), because it would possibly require more than one pole of rotation (Dewey et al., 1973; Barrier and Vrielynck, 2008; Jagoutz et al., 2015; Hinsbergen et al., 2020).

**Variscan orogeny deduced from Carboniferous-Permian zircon ages:** The current MS attributes Carboniferous-Permian zircon ages to Variscan orogeny. That inference is not warranted because the Arabian plate was not affected by Variscan orogeny. Why not also considering other possible nearby source regions for such age-range zircons? Also, we should remember that large areas of the Arabian subcontinent is buried under Mesozoic and Cenozoic sediments. Investigated alternatives include rift magmatism of Early Carboniferous age in Israel (Golan et al., 2017. International Geology Review), buried late Paleozoic crust beneath northern Arabia (Stern et al., 2014. EPSL), continental arc magmatic rocks in Turkey correlated with southward subduction of Paleotethys (Candan et al., 2016. Tectonophysics) and continental rift granitoids in Iran linked with Neotethys opening (Jamei et al., 2020. International Geology Review).

Obviously Variscan orogeny is not the subject matter of this MS and can be avoided safely considering the doubts that surround its application to the Arabian Plate.

* The Variscan-related rock is considered as a potential source for the detrital zircons. This does not necessarily mean that the Arabian plate was affected by the Variscan orogeny. Thanks for the references! To avoid any ambiguities, "N Gondwana" has been added next to Variscan in the legend of the detrital zircon age components (Fig. 7). The age component (380-270 Ma) could point to a source area situated in the N Gondwana-related rocks, particularly for the pre-Zagros strata. It could also indicate that the source area was located in Variscan-related as well as in N Gondwana-related rocks for the younger strata.

A new text has been added as follows: The late Paleozoic age components from the pre-Zagros formations are likely Gondwana-related, unlike the comparable age components from the younger formations that likely involve Variscan-derived detritus (e.g. Barber et al., 2019).

**Stratigraphic chart missing:** Presentation of an uninterpreted stratigraphy is essential for this paper. Figure 12 presents the stratigraphy but it 1) comes at the end and 2) is interpreted to go along with the proposed model. A simple stratigraphic chart to go with section 2 would be very helpful for the readers of this paper especially if the reader is unfamiliar with the region.

* Please look at page 14 of the early version of the manuscript to see the stratigraphic chart.

**Some detailed comments:**

**L102**: Here, the phrase "the basin shallowed upward" doesn't make sense. Shallowing and deepening are used for a sequence. You mean that the basin shallowed over time. Therefore, the sentence should be rearranged like "an upward shallowing is suggested by sedimentary facies change in the basin deposits."

* The text has been edited as below: Later during most of the Late Cretaceous, the basin was a site for further deposition, and it was filled with shelfal and lagaoonal carbonates.

**Figure 12:** Events shown by circled numbers 1-4 are not explained in the figure caption. Also, there is no reference to this figure in the text to help the reader about those numbers.

* Actually the numbers are explained in the caption of Fig. 12, but to be more precise, a new text was introduced to the sentence as below: Numbered arrows represent estimated relative timing of thrusting **of their respective terranes**

**Figure 13:** Below are some questions

**Panel a (Late Cretaceous time):** Two subduction zones are shown one underneath the Sanandaj-Sirjan zone, and one is intraoceanic. Neither is associated with magmatic activity. Any explanation? **Panel b (Paleocene):** The intraoceanic subduction is still amagmatic. No explanation?

* The point of Panel A was to highlight the obduction of the ophiolitic terrane onto the Arabian plate, particularly the period after the intraoceanic arc magmatism. There could have been some magmatism associated with the downgoing slab afterward, but lacking evidence for a well-developed magmatism in the upper plate, drawing it for a downgoing slab deemed unnecessary. However, a sketch showing the generation of magma might be better for the slab

beneath the Sanandaj-Sirjan zone, but as mentioned the point was to highlight the activity near Arabia, and that is also the focus of the related text in the figure. In general, there could be several reasons for lacking well-developed magmatism such as: (i) a relatively fast obduction, (ii) a flat-slab subduction, or (iii) a high-angle subduction, which might be the case due to the anticipated slab breakoff. To make the steps more clear, the Fig. 13 has been updated to be more consistent with the discussed the proto-Zagros and Neogene Zagros foreland basin.

**Panel c (Eocene):**
(1) Arc magmatism over the intraoceanic subduction (closer to the Arabian margin). What volcanic rock formation is produced here?
* These magmatic activities depict the Eocene Intrusions within the Upper Cretaceous ophiolitic terranes (e.g. Aswad et al., 2016; Ismail et al., 2020).

(2) Slab flattening shuts off arc magmatism along the Sanandaj-Sirjan zone, but activity continues closer to the trench along the WNK. How do you explain this improbable situation?
* The point was to highlight the migration of volcanism from the Sanandaj-Sirjan zone toward the Urumieh-Dohktar magmatic arc, but to improve the model, magmatism has been added and the related documented intrusions were added as well (e.g. Moritz et al., 2006; Abdollahi et al., 2020).